# Imputation of ancient human genomes

**Bárbara Sousa da Mota[1,2], Simone Rubinacci [1,2], Diana Ivette Cruz Dávalos [1,2], Carlos Eduardo G. Amorim[3], Martin Sikora [4], Niels N. Johannsen[5], Marzena H. Szmyt[6], Piotr Włodarczak [7], Anita Szczepanek[7,8], Marcin M. Przybyła[9], Hannes Schroeder[10], Morten E. Allentoft[4,11], Eske Willerslev[4,12,13,14], Anna-Sapfo Malaspinas[1,2,15] ✉ & Olivier Delaneau [1,2,15] ✉**

Due to *postmortem* DNA degradation and microbial colonization, most ancient genomes have low depth of coverage, hindering genotype calling. Genotype imputation can improve genotyping accuracy for low-coverage genomes. However, it is unknown how accurate ancient DNA imputation is and whether imputation introduces bias to downstream analyses. Here we re-sequence an ancient trio (mother, father, son) and downsample and impute a total of 43 ancient genomes, including 42 high-coverage (above 10x) genomes. We assess imputation accuracy across ancestries, time, depth of coverage, and sequencing technology. We find that ancient and modern DNA imputation accuracies are comparable. When downsampled at 1x, 36 of the 42 genomes are imputed with low error rates (below 5%) while African genomes have higher error rates. We validate imputation and phasing results using the ancient trio data and an orthogonal approach based on Mendel's rules of inheritance. We further compare the downstream analysis results between imputed and high-coverage genomes, notably principal component analysis, genetic clustering, and runs of homozygosity, observing similar results starting from 0.5x coverage, except for the African genomes. These results suggest that, for most populations and depths of coverage as low as 0.5x, imputation is a reliable method that can improve ancient DNA studies.

Ancient DNA (aDNA) is characterized by pervasive *postmortem* damage, including fragmentation and deamination[1]. Moreover, the ubiquitous microbial contamination gives rise to, in most cases, low amounts of endogenous DNA, whereas contamination with DNA from related species is an even bigger challenge, as the endogenous and the contaminant DNA cannot be easily deconvolved, and thus highly contaminated genomes are often discarded from the analyses[2]. As a result, most ancient genomes have low breadth and depth of coverage, hindering confident genotype calling. Instead, pseudo-haploid data are commonly generated by sampling one allele per variant site[3,4]. Evermore methods and tools are developed to study population structure, including diploid genetic properties

[1]Department of Computational Biology, University of Lausanne, Lausanne, Switzerland. [2]Swiss Institute of Bioinformatics, University of Lausanne, Lausanne, Switzerland. [3]Department of Biology, California State University, Northridge, California, USA. [4]Lundbeck Foundation GeoGenetics Centre, Globe Institute, University of Copenhagen, Copenhagen, Denmark. [5]Department of Archaeology and Heritage Studies, Aarhus University, Aarhus, Denmark. [6]Institute for Eastern Research, Adam Mickiewicz University in Poznań, Poznań, Poland. [7]Institute of Archaeology and Ethnology, Polish Academy of Sciences, Kraków, Poland. [8]Department of Anatomy, Jagiellonian University, Medical College, Kraków, Poland. [9]Institute of Archaeology, Jagiellonian University, Kraków, Poland. [10]The Globe Institute, Faculty of Health and Medical Sciences, University of Copenhagen, Copenhagen, Denmark. [11]Trace and Environmental DNA (TrEnD) Laboratory, School of Molecular and Life Science, Curtin University, Bentley, WA, Australia. [12]GeoGenetics Group, Department of Zoology, University of Cambridge, Cambridge, UK. [13]Wellcome Sanger Institute, Wellcome Genome Campus, Cambridge, UK. [14]MARUM, University of Bremen, Bremen, Germany. [15]These authors contributed equally: Anna-Sapfo Malaspinas, Olivier Delaneau. ✉e-mail: annasapfo.malaspinas@unil.ch; olivier.delaneau@unil.ch

such as runs of homozygosity (ROH)[5], using pseudo-haploid data. However, on the one hand, methods designed for diploid and haplotypic data cannot be easily applied to pseudo-haploid data, and, on the other hand, these data come with increased bias toward the reference genome[6].

One alternative to downsampling the data to pseudo-haploid is to impute low-coverage ancient genomes. The goal of imputation is to infer missing sites, usually by using reference panels of haplotypes. Most imputation tools employ a hidden Markov model (HMM) that determines which assembly of reference haplotype chunks represents the target best. The Li and Stephen model of linkage disequilibrium (LD)[7] and haplotype sharing is at the core of this HMM. This model describes LD in terms of the subjacent recombination rates. In particular, it estimates the probability of observing a chromosome (or haplotype) given the previously sampled haplotypes from a population by considering the new haplotype as a copy of different parts of the sampled haplotypes while allowing mutations to arise. The transition rate between copying haplotypes is proportional to the recombination rate and it decreases with the number of available haplotypes to copy from.

SNP-array imputation is applied when genomes are genotyped at a subset of variant sites[8]. SNP-array imputation of modern DNA is often implemented to increase sample sizes for genome wide association studies, so as to reduce sequencing costs[9]. It is also possible to impute low-coverage genomes whose genotypes cannot be determined with certainty, in which case genotype uncertainty is captured by likelihoods[10–15]. This second type of method is suitable to impute low-coverage ancient genomes. Present-day genotypes have been imputed with increasing accuracy due to improved imputation methods on the one hand, and increased reference panel size and diversity on the other hand, such as the Haplotype Reference Consortium (HRC)[16], the 1000 Genomes Project[17] and TOPMed[18]. These advances have also been exploited to impute low-coverage ancient genomes, using present-day haplotypes, assuming matching ancestry (e.g., Martiniano et al.[19] Haber et al.[20] Saupe et al.[21] Clemente et al.[22], Cox et al.[23] Allentoft et al.[24]).

However, aDNA introduces extra challenges, including damage and potential contamination[25], and it is not clear whether ancient individuals' ancestries are well captured by reference panels of present-day individuals. Moreover, a precise quantification of possible imputation biases and errors is lacking. Hui et al.[26] proposed a two-step imputation pipeline to be applied to ancient genomes. This pipeline first imputes based on genotype likelihoods using Beagle4.1[11], and then removes sites based on their maximum genotype probability (GP), a measure of how likely each possible genotype at a site is after imputation. The resulting genotype calls are again imputed with Beagle5[27], followed by a final GP filtering step. When compared to the first imputation step alone, this pipeline yielded larger proportions of heterozygous sites that pass the specified GP threshold. Nonetheless, a single downsampled ancient European genome was used to validate these results. Cox et al.[23] further compared the proposed pipeline in Hui et al. with simply imputing with Beagle4.1 and GLIMPSE, using the same ancient genome for downsampling experiments. The precision was highest with GLIMPSE, but Hui et al. pipeline yielded the highest recall. Another recent study[28] assessed the imputation of ancient genomes performance by downsampling (0.1–2.0×) and imputing genomes from five high-coverage ancient Europeans using Beagle4.0[29] and various reference panel and sample size configurations. The authors measured genotype concordance, bias towards the reference panel and compared projections of the high-coverage, low-coverage and imputed 1x data onto principal component analysis (PCA) of present-day data. Imputation accuracy improved when i) using all populations in the 1000 Genomes reference panel instead of restricting to European genotypes alone and ii) the ancient genomes were imputed simultaneously. They found no bias increase towards the

most common reference panel allele for ancient genome coverages as low as 0.75x.

These studies[23,26,28] suggest that aDNA imputation performs well under specific conditions. However, in their assessment of imputation accuracy they used a limited sample of ancient genomes (one[26] or five[28]) and of only European descent. Furthermore, more accurate and efficient low-coverage imputation methods are available, e.g., GLIMPSE[13], than the methods they tested, i.e., Beagle4.0 and 4.1.

Here, we make use of 43 ancient genomes[24,30–50], including an ancient trio and 42 high-coverage (>10x) genomes, from four different continents and different time spans to assess i) imputation accuracy of low-coverage ancient genomes and ii) how imputation affects downstream analyses. Our overall goal is to give users a sense of the performance of imputation and to measure whether large biases are introduced in standard downstream population genetic analyses; we do not compare the pseudo-haploid and the imputation strategies in this study as we believe they can be used in a complementary fashion. To this end, we downsampled to low coverage this diverse dataset of ancient genomes, which allowed us to quantify imputation performance across different ancestries, unlike, to our knowledge, any other previous study. We imputed the downsampled ancient genomes with GLIMPSE[13], a state-of-the-art imputation and phasing tool that was shown to accurately impute low-coverage present-day genomes when relying on 1000 Genomes[17] as a reference panel. In the next sections, we show that imputation yields accurate genotypes at common variants (minor allele frequency (MAF) > 5%) starting at 0.5x, and transition and transversion sites are imputed with similar accuracy. We obtain low error imputation error rates for 1x non-African ancient genomes and we observe a decrease in imputation accuracy at rare variants for the ancient genomes dated back more than 30,000 years before present. We further assess imputation and phasing performance in the case of the ancient trio. We test different post-imputation filtering stringency levels and we find that more stringent filtering resulted in a higher number of lost alternative-allele variant sites. We show that imputation of in-solution capture (1240 K) sequenced genomes produces more accurate genotypes at the capture sites, with a small reduction in accuracy at the non-targeted common variants. To address our second goal, we study the effects of imputation not only on PCA, but also on genetic clustering and ROH analyses. Comparing to the high-coverage genomes, we obtain similar results for these downstream applications when depth of coverage is at least 0.5x.

## Results

The approach we followed in this study is schematically described in Fig. 1a. We generated two datasets: imputed genotypes from downsampled genomes and corresponding validation genotypes called from the high-coverage ancient genomes, that is, the ground truth. We started by sampling fractions of the sequencing reads from the 43 ancient genomes to obtain genomes with average depths of coverage between 0.1x and 2.0x. Then, using bcftools[51] (see Supplementary Note 1 and Supplementary Fig. 1 on the choice of genotype caller prior to imputation), we generated genotype likelihoods at biallelic sites of the 1000 Genomes phase 3 v5 data[17] phased with TOPMed[18], the imputation reference panel, including all transition sites, in contrast to other studies[28]. We then imputed the data with GLIMPSE with the different steps described in the methods section. Lastly, we called genotypes for the high-coverage genomes and filtered out low-quality calls (methods, Supplementary Note 2 and Supplementary Fig. 2), thus reducing the deamination impact. Finally, we assessed imputation performance and compared downstream analyses' results.

Three out of the 43 ancient genomes in this study constitute a trio (mother, father and son) that were re-sequenced in this study[24,47], in contrast to the remaining 40 genomes. These 43 ancient genomes were published in different studies and relate to different epochs and continents. In total, the data includes 22 individuals from Europe, five

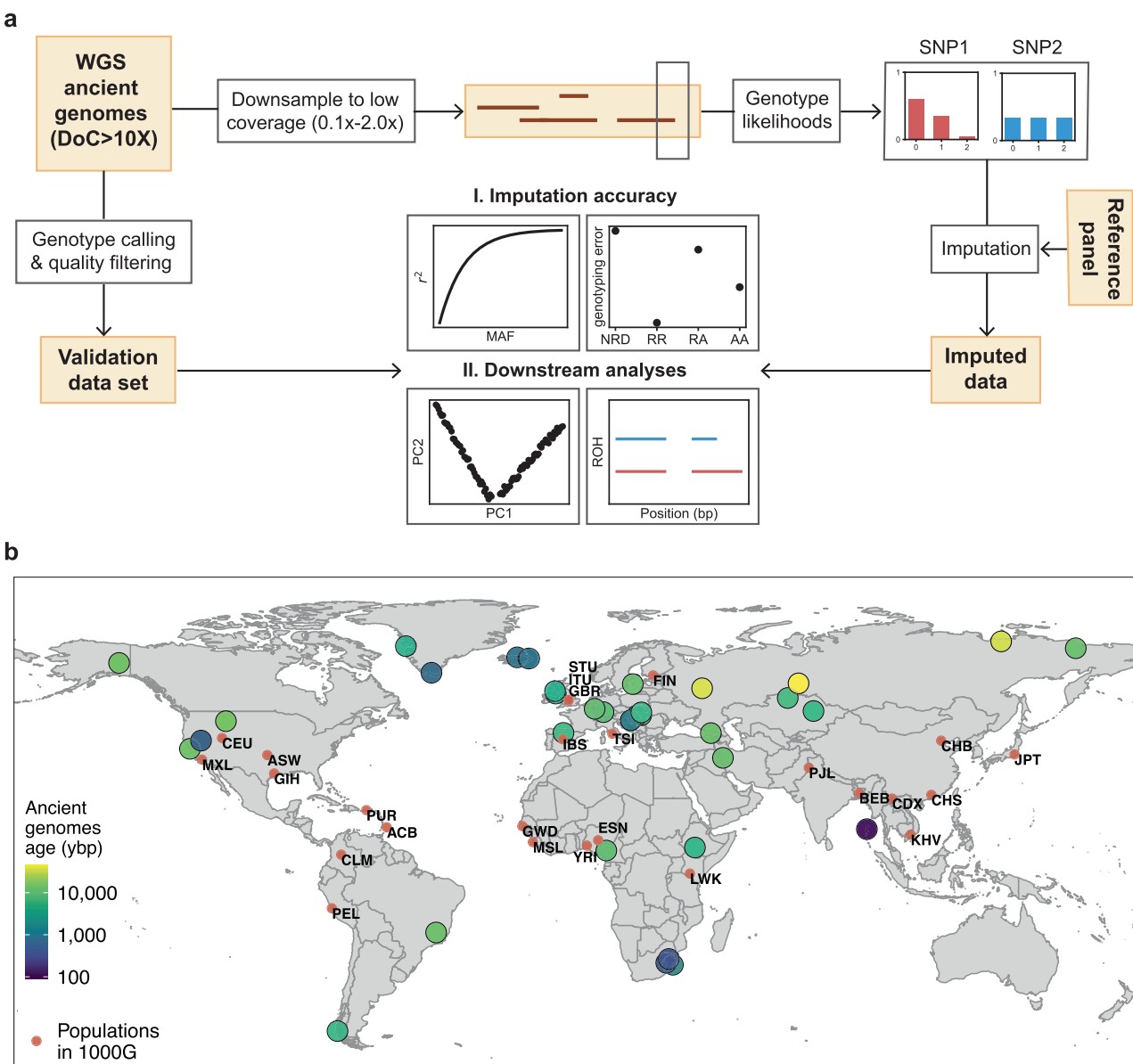

**Fig. 1 | Methodology and individual samples' origin and age. a** Overview of the procedure we followed., **b** Geographical origin and age in years before present (ybp) of the 43 individual samples used in this study as well as the different populations represented in the 1000 Genomes reference panel (ACB: African Caribbean in Barbados, ASW: African ancestry in Southwest USA, BEB: Bengali from Bangladesh, CDX: Chinese Dai in Xishuangbanna, China, CEU: Utah residents with Northern and Western European ancestry, CHB: Han Chinese in Beijing, China, CHS: Southern Han Chinese, CLM: Colombian in Medellin, Colombia, ESN: Esan in Nigeria, FIN: Finnish in Finland, GBR: British in England and Scotland, GIH: Gujarati

Indian from Houston, Texas, GWD: Gambian in Western Divisions in the Gambia, IBS: Iberian populations in Spain, ITU: Indian Telugu from the UK, JPT: Japanese in Tokyo, Japan, KHV: Kinh in Ho Chi Minh City, Vietnam, LWK: Luhya in Webuye, Kenya, MSL: Mende in Sierra Leone, MXL: Mexican ancestry in Los Angeles, California, PEL: Peruvian in Lima, Peru, PJL: Punjabi from Lahore, Pakistan, PUR: Puerto Rican in Puerto Rico, STU: Sri Lankan Tamil from the UK, TSI: Toscani in Italy, YRI: Yoruba in Ibadan, Nigeria). WGS: whole genome sequencing; DoC: depth of coverage; PC: principal component; ROH: runs of homozygosity. Made with Natural Earth. Free vector and raster map data @naturalearthdata.com.

from Africa, eight from Asia and eight from the Americas (Fig. 1b). For five of the individual samples, we had access to both high-coverage shotgun and capture data. Information concerning location and age of remains, and genome coverage is included in Supplementary Table 1 and Supplementary Table 4. To increase readability and to be able to summarize the results in more straightforward way, we split the individual samples into categories that reflect their geographical origin and/or the period they lived in: Africa, Americas, Prehistoric Europe, Historic Europe, Western Asia, South Asia and Siberia (Supplementary Table 2). While we refer to these categories throughout the text, we recognize, however, that these labels can be vague and are obviously not fully descriptive, as discussed in Coop[52].

## Accuracy of low-coverage ancient DNA imputation

We started by examining how imputation quality changes with average depth of coverage, and whether transversions are more accurately imputed than transitions, since the latter are affected by *postmortem* DNA deamination, i.e., C-to-T substitutions, which might wrongly increase the number of called heterozygous sites. We further compared imputation performance using two different state-of-the-art imputation methods, GLIMPSE and Beagle4.1[11], where the latter is a widely used imputation method and was also considered in ref. [26]. For that, we calculated imputation accuracy, $r^2$, that is, the squared Pearson correlation between genotype dosage in the aggregate of the 42 high-coverage and imputed datasets, as a

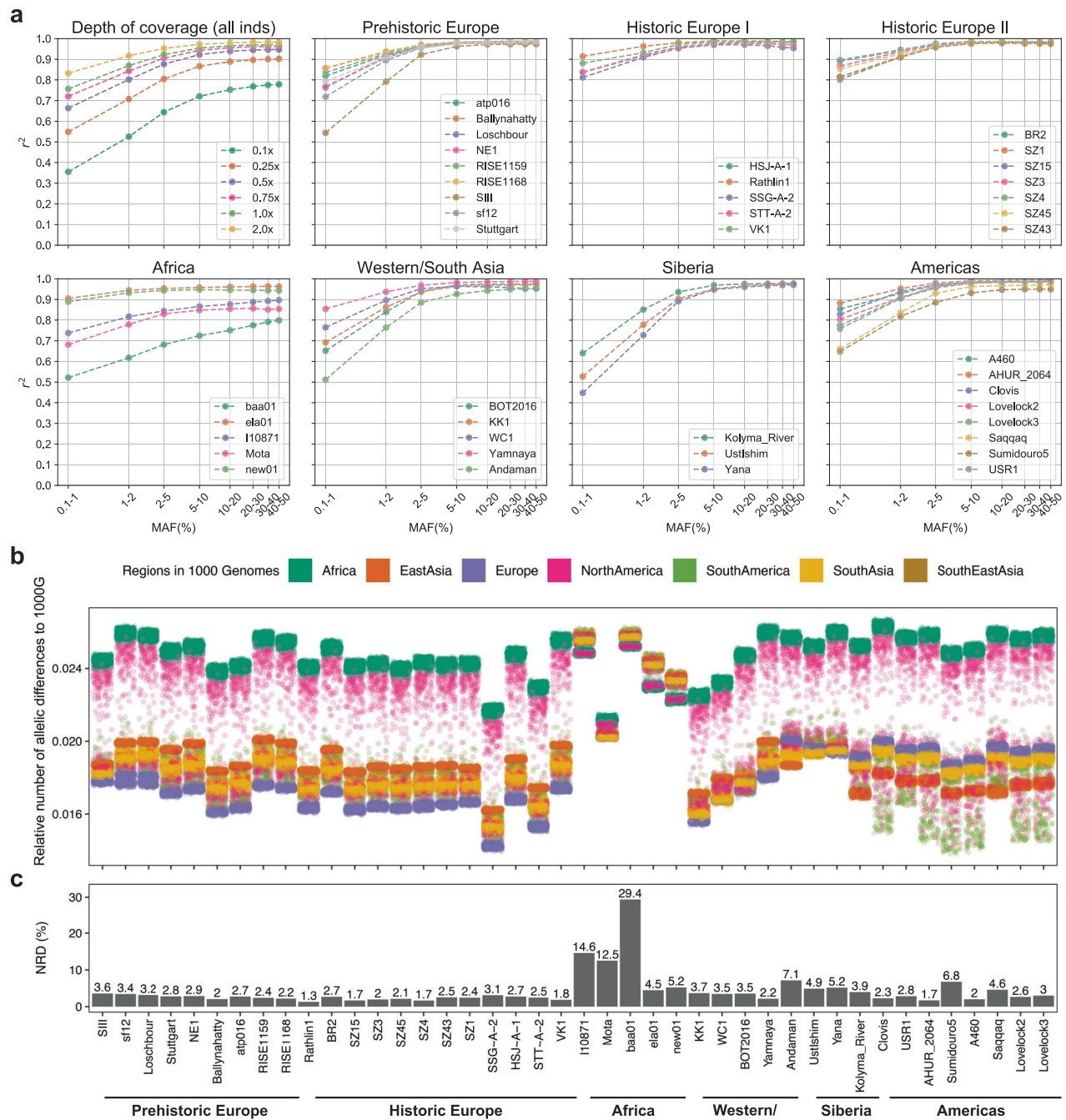

**Fig. 2 | Imputation quality assessment for 1x ancient genomes and genetic distance to 1000 Genomes reference panel. a** Imputation accuracy ($r^2$) as a function of minor allele frequency (MAF) for the 42 high-coverage genomes together downsampled to different depths of coverage (top left) and for individual 1x genomes (remaining plots). Depending on ancestry, MAF was specified from the reference populations expected to be closer to the individual in question, whenever possible, as listed in Supplementary Table 1. Individuals were put in categories that roughly reflect their place of origin and/or time. **b** Allelic pairwise differences between each ancient high-coverage genome (x-axis) and each of the 2504 individuals in 1000 Genomes reference panel, colored by continental group. **c** Resulting non-reference discordance (NRD) from imputing 42 ancient genomes downsampled to 1x. In plots **b** and **c**, individual samples are ordered by sample age within each category (oldest to the left).

function of minor allele frequency (MAF) as determined from the 1000 Genomes reference panel.

We found that imputation accuracy of ancient genomes was similar to the accuracy reported for present-day genomes when using the same imputation method (Supplementary Note 4 and Supplementary Fig. 3). Accuracy was higher at common variants (MAF ≥ 5%) (Fig. 2a), as rare variants are more challenging to impute[9,53]. Imputation accuracy was also higher for genomes with higher coverage, as these have more data. In particular, for depths equal and greater than 0.75x,

we obtained $r^2 > 0.90$ at sites with MAF > 2%, and $r^2 > 0.70$ and $r^2 > 0.95$ for rare (0.1% < MAF ≤ 1%) and common variants (MAF ≥ 10%), respectively. We then found that GLIMPSE outperformed Beagle4.1 for 1x ancient genomes, particularly at rare variants (Supplementary Fig. 4), similarly to the case of present-day genomes[13]. Finally, there were small differences in accuracy between imputed transversion and transition sites at rare variants (0.1% < MAF ≤ 1%, $r^2 = 0.75$ and $r^2 = 0.77$ for transitions and transversions, respectively), but these differences disappeared for more common variants (Supplementary Fig. 4).

Fixing depth of coverage at 1x, we evaluated how imputation performs across the 42 high-coverage genomes of different ancestries and times. In addition to imputation accuracy as a function of MAF, we quantified genotyping error rates for homozygous reference and alternative allele and heterozygous sites. We also report the non-reference discordance (NRD), that is, the ratio of the number of incorrectly imputed sites and the total number of imputed sites, excluding correctly imputed homozygous reference allele sites.

The imputation of European, Western Asian, and most Native American genomes yielded similar accuracy curves starting with lower values for rare variants ($0.5 < r^2 \leq 0.9$) and converging to $r^2 \geq 0.90$ from MAF $\geq 2\%$ (Fig. 2a). The African ancient genomes were the least accurately imputed with only two out of five imputed genomes reaching $r^2 > 0.90$, and error rates as high as 18% at heterozygous sites (Supplementary Fig. 5), the most challenging to impute, and NRD between 4% and 29% (Fig. 2c). In contrast, most non-African imputed genomes yielded NRD rates below 5%. This difference in imputation performance is likely due to underrepresentation of the different African populations in the reference panel. Indeed, the ancient African individuals in this study have much larger pairwise genetic distances to the reference panel than non-African individuals (Fig. 2b). Although the 1000 Genomes reference panel contains individuals of African origin, mostly from West Africa (Mende Sierra Leone (MSL), Gambian Mandinka (GWD), Esan Nigeria (ESN), Yoruba (YRI) and Luhya Kenya (LWK)), the genetic diversity in Africa[54] is not well represented in this panel. And yet, Native American genomes were also accurately imputed, even though the populations in the reference panel show different admixture moieties, ranging from low (e.g., Puerto Rican (PUR)) to high Native American (e.g., Peruvian (PEL)) admixture proportions[17]. In fact, Fig. 2b shows that the South American reference individuals tend to be genetically close to the Native American genomes (small pairwise allelic differences). We further confirmed the contribution of reference haplotypes from the Americas to the imputation of ancient Native American genomes by removing one continental group at a time from the reference panel. We found that imputation performance was only affected when using a reference panel without the American populations (Supplementary Note 7 and Supplementary Fig. 6). Imputation accuracy dropped to 0.46 from 0.78 at variants in the lowest MAF bin (0.1–1.0%), while it was only slightly smaller (-0.97 vs. -0.98) at common variants (MAF > 5%).

Sample age is also expected to affect imputation performance, as long-time distances to the present could translate into large coalescent times between the reference populations and the ancient individuals[55]. While overall imputation performance seems to be unaffected by sample age (Fig. 2c), imputation accuracy at rare variants (MAF < 2%) is considerably low for the three oldest individuals, i.e., Yana (~32,000 ybp)[40], SIII (~34,000 ybp)[48], and Ust'Ishim (~45,000 ybp)[39], as shown in Fig. 2a. We found significant (at 5% threshold) negative correlations between sample age and imputation accuracy at the two lowest MAF bins, i.e., 0.1–1% and 1%–2%: $r_s = -0.465$ ($p$ value = 0.003) and $r_s = -0.324$ ($p$ value = 0.033), respectively (Supplementary Note 8 and Supplementary Fig. 7).

The newly re-sequenced ancient trio (mother, father, son) allowed us to use an orthogonal approach based on Mendel's rules of inheritance to measure imputation and phasing quality. This trio was sampled in a Late Neolithic mass burial at Koszyce[24,47] and was re-sequenced in our study to a depth of coverage of 27.5x (mother, RISE1159), 18.9x (father, RISE1168), 5.4x (son, RISE1160). In this analysis, imputation errors corresponded to sites where parental and offspring genotypes disagreed with Mendel transmission rules. Here, we excluded sites that are homozygous for the reference allele in the three genomes as these positions are easier to impute. We estimated phasing accuracy in terms of switch error rate. The switch error rate is assessed for every two consecutive heterozygous sites by verifying if the alleles for the two sites are located on the correct haplotypes following the

expected configuration from the trio. Mendel error rates ranged from 1.3% at 4x to 12.2% at 0.1x (Fig. 3a). For 1x data, in particular, Mendel error rates were between 1.5% and 2.9% across the 22 autosomes. These error rates agree with previously estimated imputation errors (Fig. 2c and Supplementary Fig. 5). Switch error rates varied between 1.6% at 4.0x and 8.2% at 0.1x, with errors for 1x data in the range 1.6–3.0% (Fig. 3b). For present-day genomes and small sample sizes, switch error rates are typically between 1% and 5%[56–58], and we achieved similar accuracy when imputing and phasing the genomes downsampled to a minimum coverage of 0.25x.

After imputation, we can filter the data based on the maximum genotype probabilities (GP) for a site. GP is a measure of how likely each genotype is to be true and takes values between 0 and 1 that sum to 1 across the possible genotypes. To determine which GP value we would use to filter the imputed data prior to downstream analyses, we applied GP filters starting at 0.70 and up to 0.99 to four different imputed ancient genomes downsampled to 0.1x and 1.0x (RISE1168[24,47], SIII[48], Ust'-Ishim[39] and Mota[34]). We then quantified imputation accuracy and genotype discordance. We observed a greater boost in accuracy as the GP filter becomes stricter for 0.1x imputed data than for 1x data (Fig. 4a). In the case of 1x data, accuracy slightly improved for sites with MAF > 5%. The exception was the individual sample Mota (Africa), where the gain in accuracy for a specific GP filter had similar magnitude across sites with different MAF values. This African genome yielded the second lowest imputation accuracy amongst the 42 ancient high-coverage genomes downsampled and imputed in this study. Genotype discordance followed the same trend (Fig. 4b). Genotyping error rates were higher for 0.1x than for 1x imputed genomes, for whom error rates remained below 5%, except for Mota. Increasing GP filtering values decreased these error rates in all instances. Then, we looked at how GP filtering affects the number of correctly imputed heterozygous sites (Fig. 4c). The proportion of lost heterozygous sites was much higher in the case of 0.1x data, explained by the lower imputation accuracy for this coverage. For 0.1x data, filtering out sites with GP < 0.70 removed around 15% of correct heterozygous sites in the least. When GP ≥ 0.99, only between 20% and 43% of correct heterozygous sites remained. In contrast, the imputed 1.0x genomes lost a small fraction of their heterozygous sites as stricter GP filters were applied. This fraction was smallest amongst the genomes of European ancestry (<8%, RISE1168 and SIII) and largest for Mota (22%), a reflection of how accurately these genomes were imputed. In the end, a trade-off must be made between loss of heterozygous sites and imputation accuracy. Based on these results, we chose to remove sites with MAF < 5% and set to missing imputed sites with GP < 0.80, for most of the downstream analyses, thus keeping most heterozygous sites for 0.1x data while controlling for imputation accuracy.

Ancient DNA studies often resort to hybridization-capture sequencing, that increases the depth of coverage at captured pre-specified sites[59–62]. Capture data using the widely used 1240K array[63–65] were previously generated for five of the 42 high-coverage ancient genomes, with depths of coverage at the capture sites between 1.5x and 11.1x (Supplementary Table 4). We found that imputation accuracy was higher at the intersection of 1240 K and 1000 Genomes sites at variants with MAF < 5%, but common variants were imputed with similar accuracy at the capture sites and outside of these (Fig. 5 and Supplementary Fig. 8). To study the effect of depth of coverage on imputation of capture genomes, we imputed downsampled genomes with depths of coverage between 0.1x and 2.0x and 0.1x and 5.0x at the capture sites for BOT2016 and I10871 and Stuttgart, respectively. Imputation accuracy reached 0.90 only at 2x at common variants (MAF > 5%) for BOT2016 and Stuttgart, whereas imputation performance was lower for I10871, an African individual (Fig. 5). For the Stuttgart genome, the gain in imputation accuracy was small when increasing depth of coverage from 4.0x to 5.0x ($r^2 \geq 0.94$ to $r^2 \geq 0.95$

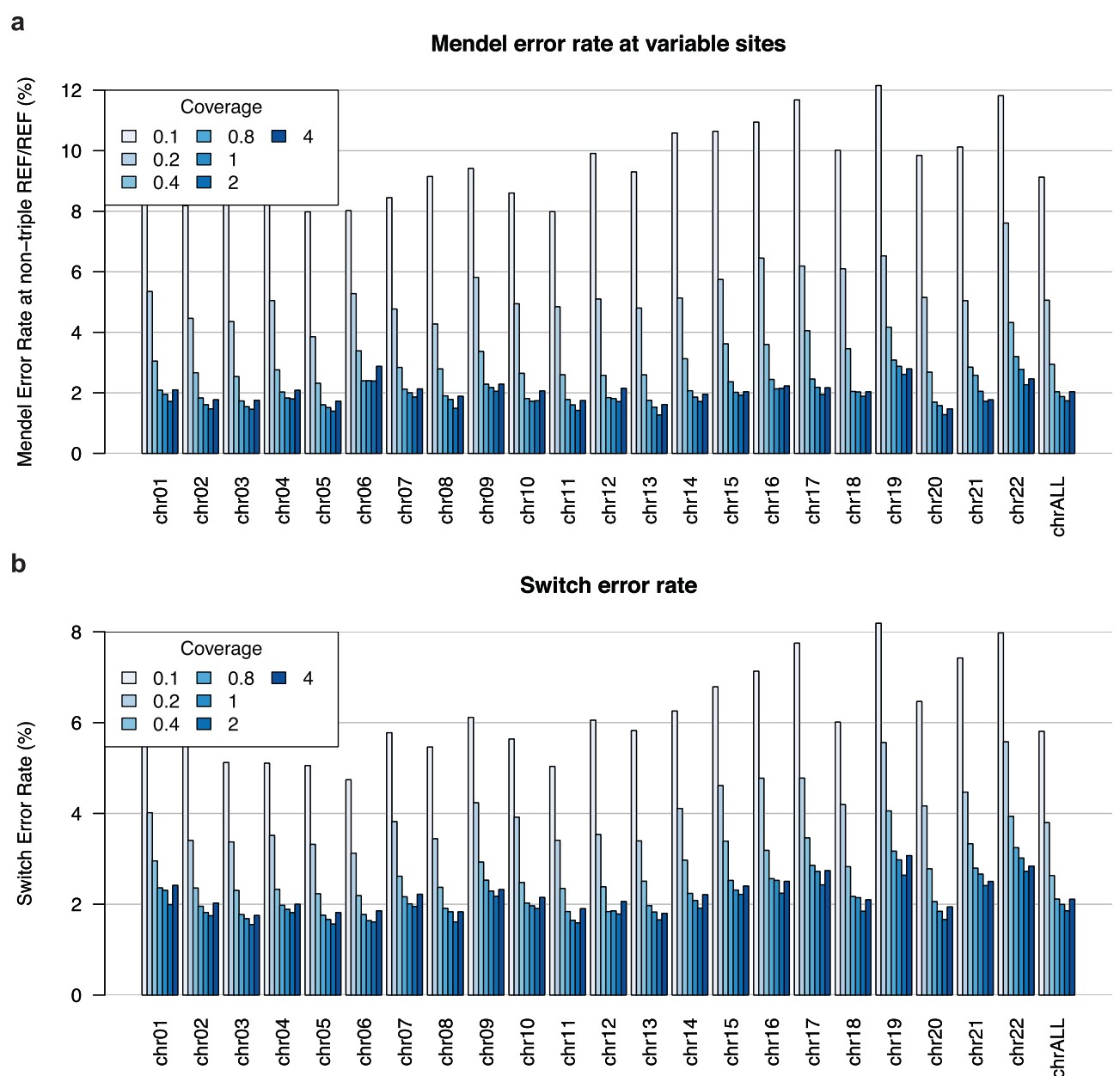

**Fig. 3 | Imputation and phasing accuracy for the Koszyce trio. a** Mendel error rate across the 22 autosomes is counted when the parental and offspring genotypes violate Mendel transmission rules, excluding sites at which all three non-imputed genomes are REF/REF. **b** Switch error rates averaged over the three genomes. A switch error is counted between two consecutive heterozygous genotypes when the reported haplotypes are not consistent with those derived from the trio.

for MAF > 5%). Moreover, for the same individual samples, the imputation performances of 1x capture and shotgun-sequenced data with depth of coverage between 0.1x and 0.5x were equivalent (Supplementary Fig. 9).

**Imputation effect on downstream analyses**

In order to detect and quantify potential bias introduced by imputation, we compared the results of downstream analyses, namely, principal component analysis (PCA) and genetic clustering analyses, performed with the high-coverage and imputed genomes, after filtering for MAF and GP (imputed data). These methods are broadly used in population genetics to investigate population structure and demography. PCA is a dimension reduction technique that helps visualize patterns of population structure. In the genetic clustering analyses, ancestries are estimated as the sum of K different clusters determined from the data in an unsupervised fashion. We further explore the

potential of imputing low-coverage ancient genomes by estimating ROH, whose classical applications require diploid data. ROH segments are unbroken homozygous regions of the genome that contain information about past and recent breeding patterns[66]. ROH have been found in all populations, but their number and size vary, depending on demographic histories.

For the PCA, we calculated the first ten principal components of the 1000G reference panel and projected both the high-coverage and corresponding imputed ancient genomes onto those. We have included both transition and transversion sites in this analysis.

Both the imputed 1x and high-coverage ancient genomes were in the expected continental groups as defined by present-day individuals in the two first principal components (Fig. 6a). They also tended to colocalize, which was particularly the case for ancient individuals clustering with present-day Europeans, suggesting limited bias is

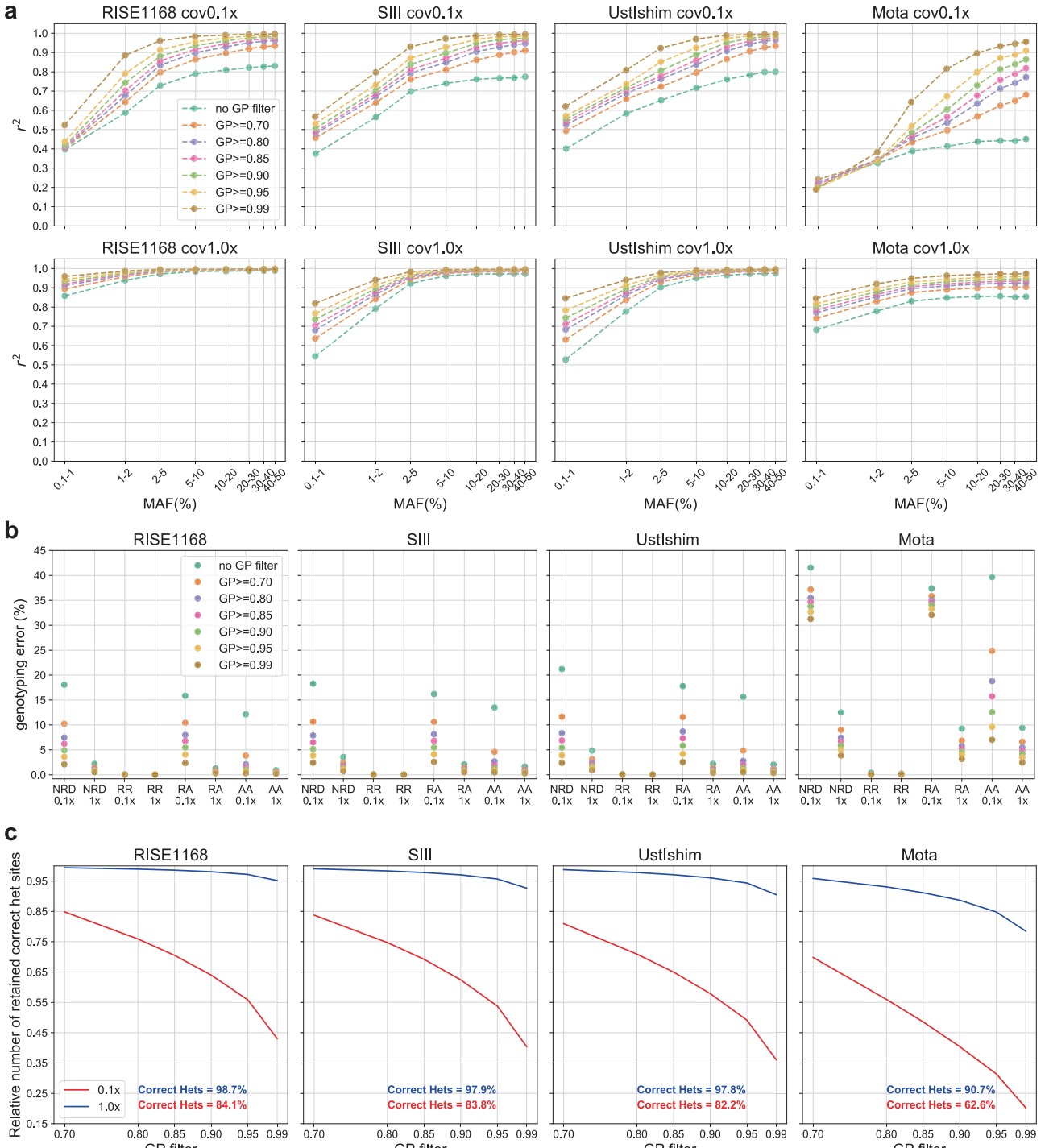

**Fig. 4 | Effects of applying different thresholds when filtering for genotype probability (GP) in the case of four imputed 1x ancient genomes (RISE1168[24,47], SIII[48], Ust'-Ishim[39] and Mota[34]). a** Imputation accuracy. b Genotype discordance between imputed and non-imputed genomes for homozygous reference allele (RR), heterozygous (RA) and homozygous alternative allele (AA) sites, and also the non-reference discordance (NRD). **c** Proportion of correctly imputed heterozygous sites retained for 0.1x and 1.0x data for each of the four genomes. The percentage of correctly imputed heterozygous sites for 0.1x and 1.0x before GP filtering are represented in red and blue, respectively, in (**c**).

introduced by imputation in the PCA results. To further verify whether imputation introduced bias in this analysis, we took the difference in coordinates between validation and corresponding imputed 1x genomes for each principal component. As shown in Fig. 6b, the normalized differences between the two datasets were small and did not deviate significantly from 0 (t-test *p* values > 0.01). Additionally, we found that only genomes with coverage as low as 0.1x and 0.25x show some significant deviation from 0 (Fig. 6c) for some principal

components, however, the imputed data were still placed in the expected continental clusters in the PCA space (Supplementary Fig. 10). This is particularly clear for European ancient genomes. These results show that the differences between imputed and high-coverage coordinates tended to be centered on 0 for the first principal components, in particular for genomes with coverage above 0.25x, suggesting that imputation did not introduce a significant bias to the PCA.

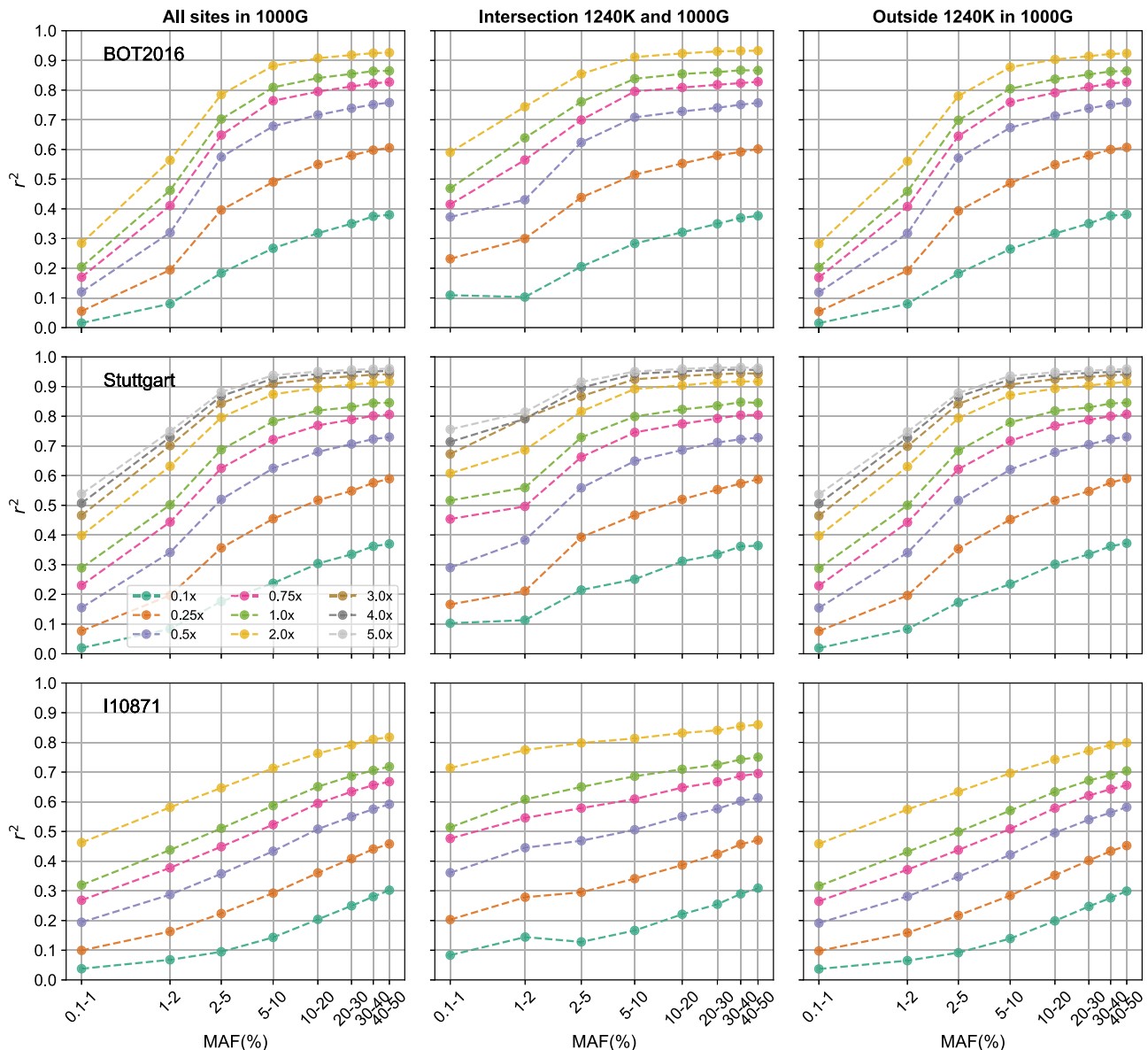

**Fig. 5 | Imputation accuracy, $r^2$, as a function of minor allele frequency (MAF) for three genomes sequenced with a 1240k capture.** From top to bottom: BOT2016, Stuttgart and I10871. We evaluated imputation accuracy at all variant sites in 1000 Genomes (first column), at the intersection of the 1240 K array and the 1000 Genomes panel (second column), and at the sites only found in 1000 Genomes (third column). The capture genomes were downsampled to coverages between 0.1x and 2.0x, as measured on the 1240 K sites.

For the genetic clustering analyses, we focused on the European genomes. Present-day Europeans can generally be modeled with three ancestral populations: western hunter-gatherers, early European farmers and Steppe pastoralists[41]. Ancient European individual samples tend to exhibit different distributions of these three ancestries across time and space. We asked whether imputation of European ancient genomes artificially increases the amount of inferred Steppe-like ancestry for these individuals, since most present-day European individuals have Steppe ancestry, including the European populations in the 1000 Genomes reference panel. For instance, we assessed whether the Steppe-like component increases in imputed western hunter-gatherer genomes like Loshbour[41]. To this aim, we performed unsupervised admixture analyses with the software ADMIXTURE[67], including transitions and transversions. We used as a reference panel the genetic data of 61 ancient individuals[65,68–71] present in the 1240K dataset[63], including nine western hunter-gatherers, 26 Anatolian farmers and 26 individuals with Steppe-like ancestry (see

Supplementary Table 5). We estimated ancestry proportions for the imputed and validation data separately varying the number of clusters ($K$) between two and five. For $K = 2$, 4, and 5, we observe qualitatively similar results for imputed and high-coverage data (see Supplementary Note 11). Here we show the results obtained with $K = 3$ (Fig. 7a), as these clusters seemingly capture the three aforementioned ancestries. The admixture proportions are qualitatively similar between the high-coverage ancient genomes and the corresponding imputed ones, and, in the particular case of Loschbour, the only western hunter-gatherer imputed in this study, we estimated 100% western hunter-gatherer-like ancestry with both imputed 1x and high-coverage data (Fig. 7b). In order to compare the admixture results across imputed data with different depths of coverage, we took the difference between ancestry proportions estimated for the validation and imputed genomes for each ancestry component and each coverage (Fig. 7c). We observed larger differences with imputed 0.1x and 0.25x data. For the remaining depths of coverage, the small differences distributed around 0 show

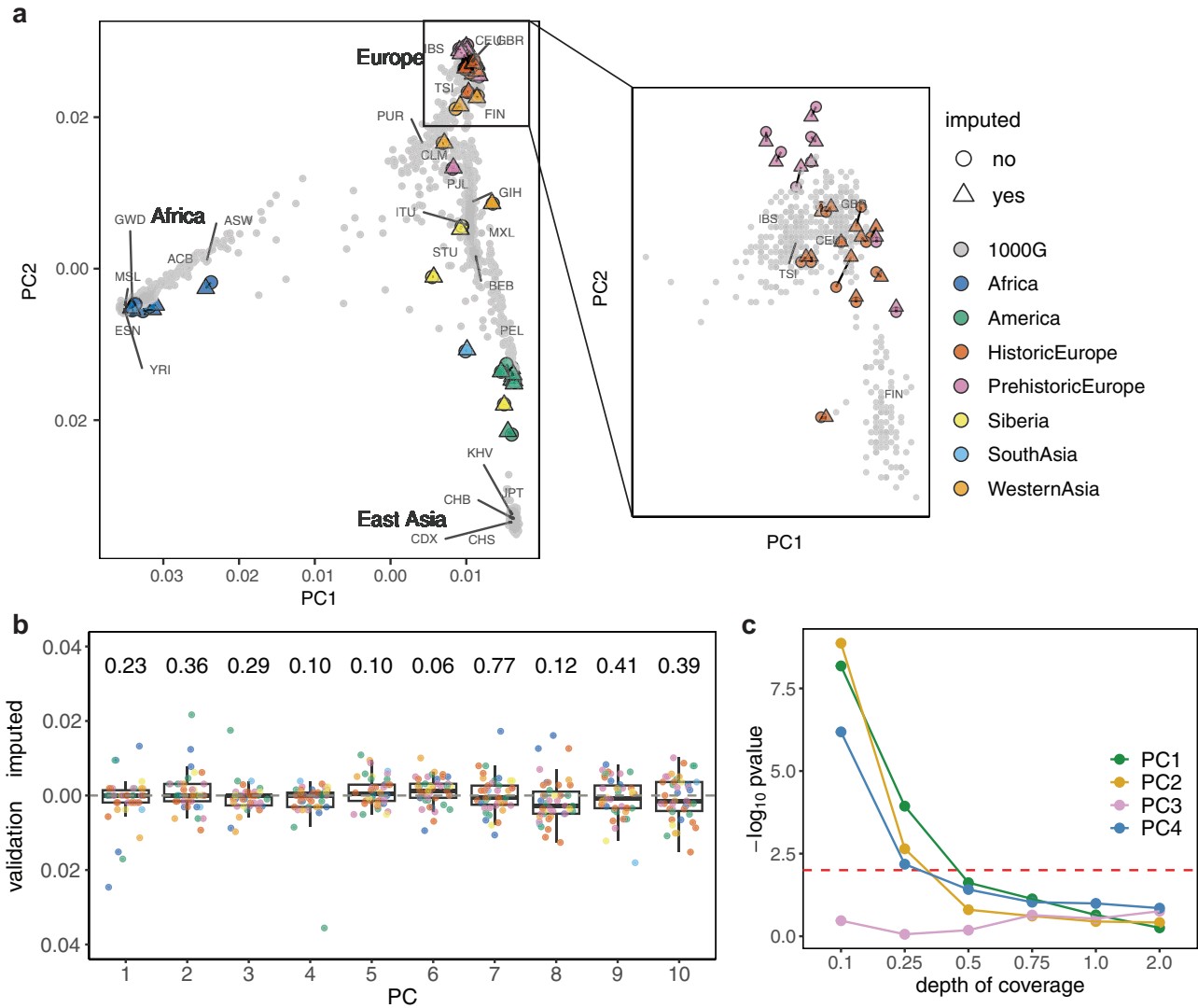

**Fig. 6 | Principal component analysis (PCA) of imputed and high-coverage ancient genetic data, and present-day data in 1000 Genomes reference panel. a** Projections for 1x imputed, high-coverage and present-day data along the first two principal components, where 1000 Genomes individuals are plotted in gray and population labels are shown in the average location of the individuals from the same population, ancient individuals are colored by region and/or epoch, with the high-coverage and imputed individuals represented by full circles and triangles, respectively; the plot on the left contains the coordinates of the whole data set and the plot on the right shows the coordinates of European modern individuals as well as of the European-labeled ancient individuals that cluster with these. **b** Boxplots (where horizontal lines represent, from bottom to top, the first quartile, the median and the third quartile, and the whiskers lengths are 1.5 times the interquartile range) of the normalized differences in coordinates between validation and corresponding 1x imputed genomes for the first 10 principal components and resulting $p$ values from testing whether differences are significantly different from 0 ($n = 42$ independent individual samples, two-sided t-test, no adjustments were made for multiple comparison); individual data points are overlaid and colored according to the region and/or epoch as in the previous plot. **c** $-\log_{10} p$ values obtained when testing whether differences between imputed and validation data are significantly different across the six depths of coverage and for the first four principal components ($n = 42$ independent individual samples, two-sided t-test, no adjustments were made for multiple comparison); the red dashed line indicates a $p$ value of 0.01.

that imputation introduced limited bias towards a particular ancestry in this analysis.

Then, we first quantified ROH using transversions only to minimize the aDNA damage impact on the validation estimates. We examined how well the imputed and the validation ROH overlapped in chromosome 10 for each depth of coverage and for four different individuals, namely Ust'-Ishim[39] (Siberia), Rathlin1[72] (Europe), A460[38] (Americas), and Mota[34] (Africa) (Fig. 8a). The imputed 0.1x data had an excess of ROH when compared to the high-coverage data. This likely results from i) reduced imputation accuracy and ii) removal of a large proportion of heterozygous sites when applying post-imputation filters (Fig. 4c). As the depth of coverage increased, the number of falsely identified ROH tended to decrease, while most validation ROH were also found amongst the imputation ROH. We then compared the total ROH

lengths, stratified by segment size, measured in the imputed data with the validation data for the different depths of coverage and the same four individuals (Fig. 8b). Again, we found the largest discrepancies between validation and imputed 0.1x data, with an excess of ROH segments, particularly of the shortest kind (0.5–1.0 Mb). For coverages above 0.1x, the total ROH lengths in the imputed genomes were close to the validation ROH, particularly for A460 (5% difference) and Ust'Ishim (0.7% difference). Lastly, restricting to imputed 1x data, we contrasted the total length of small ROH (<1.6 Mb) with the total length of longer ROH (≥1.6 Mb) obtained with transversions only (Fig. 8c) and all sites (Fig. 8d). When using transversions only, the total ROH lengths estimated for high-coverage and corresponding imputed 1x genomes were similar, particularly for the European genomes. Furthermore, the ROH trends for the ancient individuals mostly agreed with documented ROH

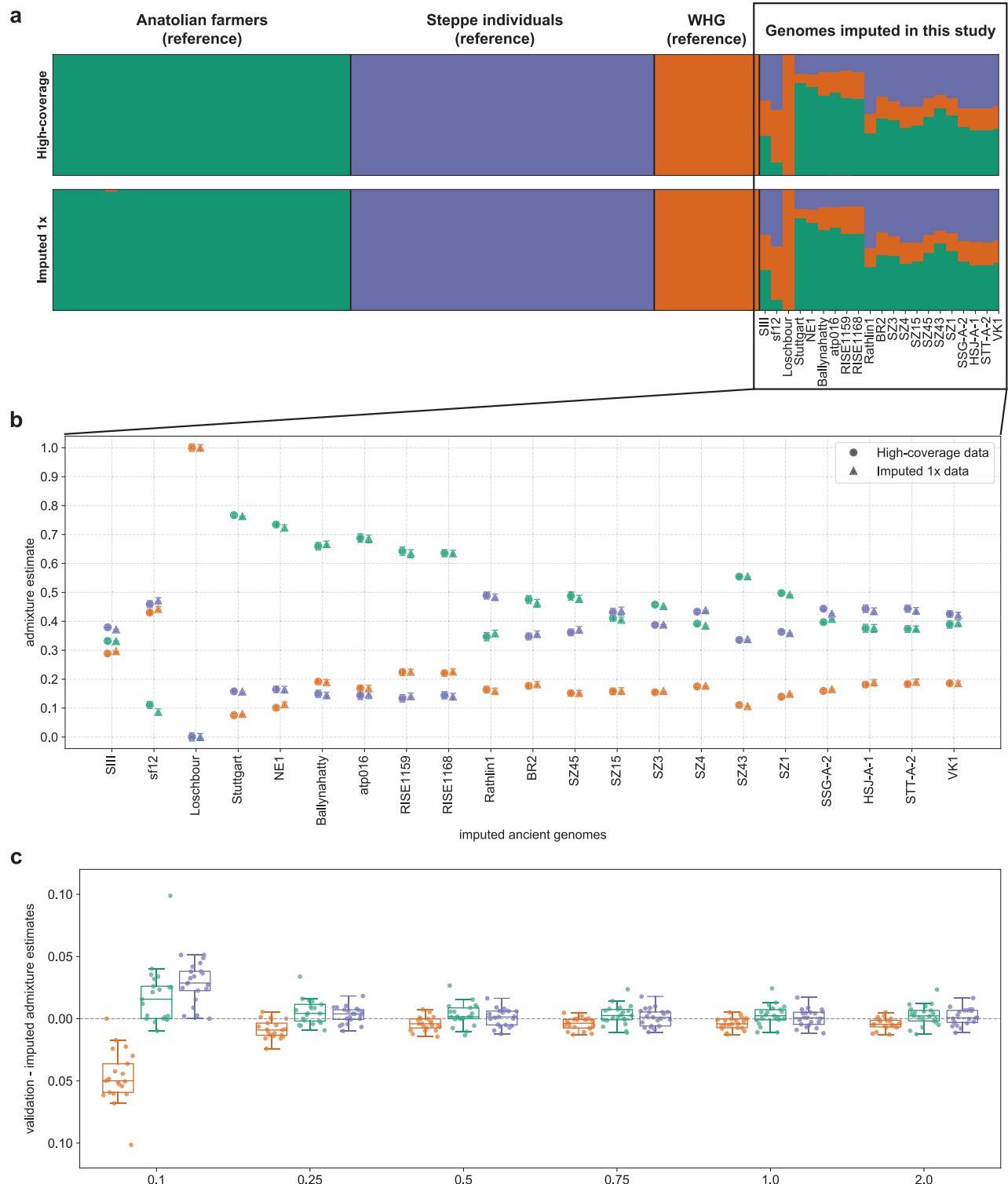

**Fig. 7 | Unsupervised admixture analyses of European ancient individuals with three clustering populations, where Anatolian farmers, Steppe individuals and Western Hunter-Gatherers (WHG) are split into the three clusters. a** Resulting admixture proportions and clusters for the reference and the 21 European individuals in this study, with validation results on top and imputed 1x below. **b** Admixture estimates for each of the three clusters obtained with imputed 1x (triangles) and validation (full circles) data for each of the 21 individuals, where error bars represent one standard error of the estimates. **c** boxplots (where horizontal lines represent, from bottom to top, the first quartile, the median and the third quartile, and the whiskers lengths are 1.5 times the interquartile range) of the differences between the values of ancestry components obtained with the high-coverage and imputed data across all depths of coverage ($n = 21$ independent European individual samples).

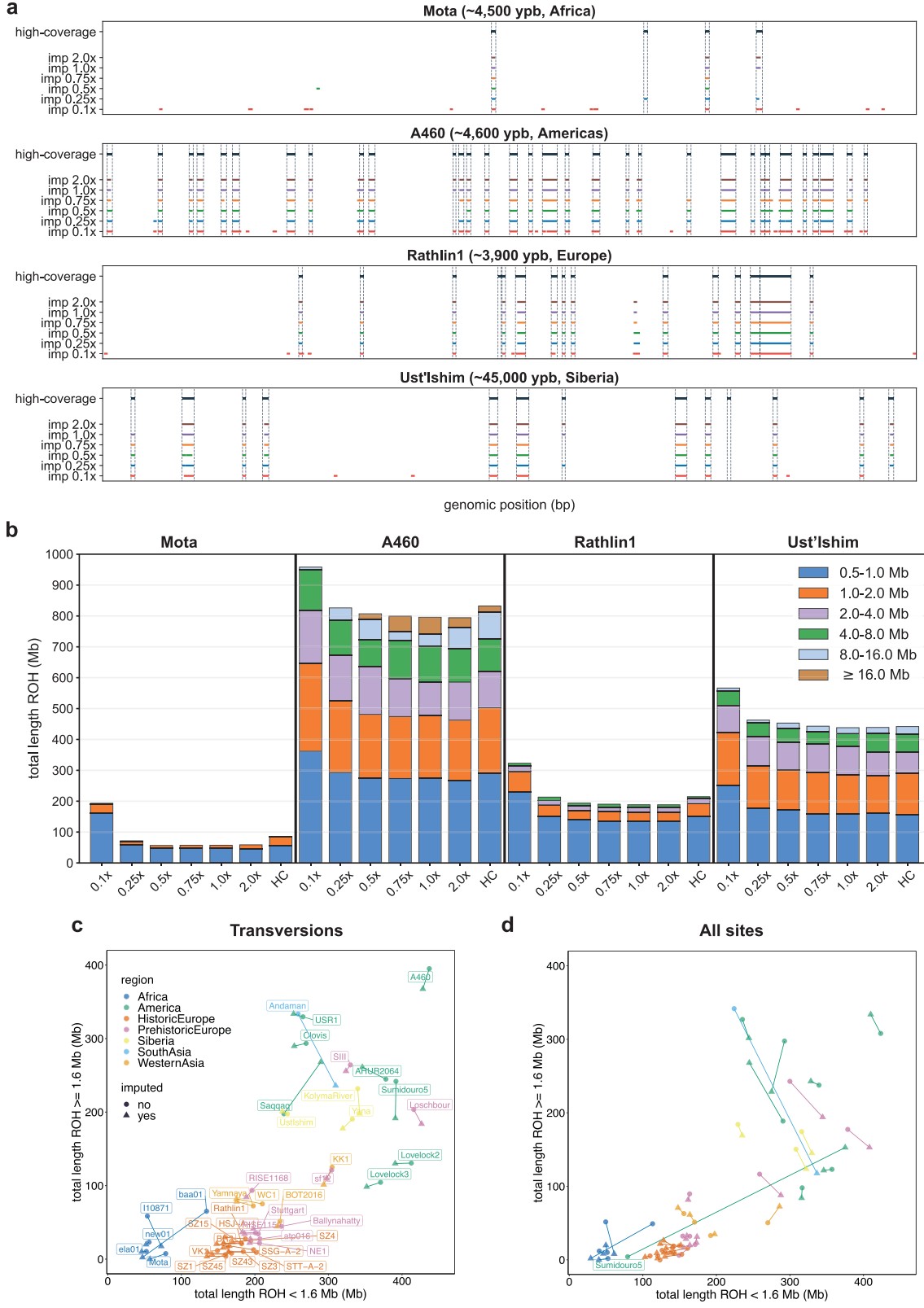

**Fig. 8 | Runs of homozygosity (ROH) estimates for the high-coverage and corresponding imputed genomes. a** ROH locations in chromosome 10 found using transversions only with high-coverage and imputed genomes, in the case of four ancient individuals, namely, Mota[34] (~4500 ypb (years before present), Africa), A460[38] (~4600 ypb, Americas), Rathlin1[73] (~3900 ypb, Europe), Ust'-Ishim[39] (~45,000 ypb, Siberia). **b** Total length of ROH discriminated by individual ROH length categories, estimated for imputed and high-coverage genomes (HC) using transversion sites for the four aforementioned individuals. **c** Total length of long (≥1.6 Mb) vs. small (<1.6 Mb) ROH segments for validation (full circles) and 1x imputed (triangles) genomes using transversion sites only and **d** using transversions and transitions.

for their present-day counterparts, with Africans having the smallest total ROH lengths and Native Americans the longest[66].

When we added transitions to estimate ROH, the distance between imputed and validation ROH increased for some genomes (Fig. 8d). In the case of the ancient Native American Sumidouro5[38], this distance dramatically increased. The high-coverage estimate for Sumidouro5 was now located between the African and European values, but the imputed estimate remained close to both the high-coverage and imputed values obtained with transversions only. For this genome, we found major differences between high-coverage ROH sizes obtained with transversions only and all sites, whereas the corresponding imputed ROH were highly consistent (Supplementary Fig. 18). This indicates that the discordance between validation and imputed ROH, when transitions were included, originated from the validation data. Indeed, Sumidouro5 is a very damaged genome (40% deamination rate at read termini)[38], which likely led to an excess of heterozygous calls in the high-coverage data, despite the quality filtering (see methods).

## Discussion

In aDNA studies, pseudo-haploid data generation is standard procedure to handle low-coverage genomes. Ancestry information can be recovered from analyses of these data, such as PCA and genetic clustering analyses which work well at coverages as low as 0.1x[73]. Compared with pseudo-haploid data, imputing genotypes allows to work on diploid genomes and to directly apply some population genetic tools developed for modern data.

Here we showed that low-coverage ancient genomes can be imputed with similar accuracy as modern genomes. In particular, for shotgun-sequenced data, we obtained accurate results at common variants, for coverages starting at 0.5x from MAF > 5% (or at 0.75x from MAF > 2%). However, we observed that this threshold is dependent on the ancient genomes' ancestry. The representation of a given population in the reference panel can have a profound impact on imputation accuracy, with genotyping errors at alternative allele sites above 5% and up to 25% among African 1x genomes. Despite the absence of 100% Native American reference populations, most Native American ancient genomes were accurately imputed. The presence of haplotypes with partial Native American ancestry in the reference panel allowed us to recover variants private to Native American populations. Moreover, we found that age can negatively impact imputation accuracy of rare variants, which was the case of three non-African individual samples older than 30,000 years. These results have far-reaching implications for the potential of imputing ancient genomes, since it is not guaranteed that there will be a present-day population that directly descends from the ancient individual's population without having admixed. Our results suggest that, on the one hand, using admixed reference populations that share recent ancestry with the ancient genomes can be enough to attain accurate imputation, even at rare variants, and, on the other hand, we can still impute common variants well in the case of non-African genomes that are either very old, such as Ust'Ishim, or that are poorly represented in the reference panel, such as Andaman, likely owing to their common history.

Furthermore, using five genomes that were both obtained via in-solution capture and shotgun sequencing (>10x for the latter), we found that imputation performance of capture-sequenced data was higher at the capture sites than outside of these and particularly for rare variants, and imputing a 1x (target sites) capture genome and a 0.25x shotgun-sequenced genome result in similar error rates. Moreover, imputation accuracy was below 0.90 for coverages below 2x at the target sites. We therefore recommend a minimum depth of coverage of 2x at capture sites, but ideally higher than that (imputation accuracy levelled off at around 4x for the Stuttgart genome), to attain accurate imputed calls, in the case of well represented ancestries.

For most genomes, we obtained similar results with high-coverage and imputed data with coverages as low as 0.5x for the downstream analyses we carried out, i.e., PCA, admixture clustering and ROH estimation. Imputation did not introduce major bias for the first principal components, nor did it considerably increase the proportion of any of the three main ancestry components found in Europeans. The similarity of validation and imputed ROH segments is worthy of note, since ROH estimation typically requires reliable knowledge of genotypes, which is only available for high-coverage genomes. This means that ROH estimation methods designed for diploid data can be applied to low-coverage ancient genomes after imputation.

Although we did not remove transition sites prior to imputation, we found that transversion and transition sites were imputed with comparable accuracy. In fact, when we compared ROH estimates performed with transversions and all sites, we observed that imputation corrected ROH in the case of Sumidouro5, with 40% C-to-T mismatch frequency at the end of the reads. Given this observation, imputation of ancient genomes has the potential of correcting genotypes that are affected by damage and other sources of error. Whether imputation can help reducing the effect of contamination remains to be assessed.

We did not explore numerous genotype and haplotype-based applications that can greatly benefit from imputation of low-coverage ancient genomes, such as temporal selection scans and local ancestry inference. Moreover, genotype imputation, in general, is expected to improve as more and larger reference datasets become available. The recent release of 200K whole-genome sequences in the UK Biobank[74], which can be used as a reference panel for imputation, offers an opportunity to improve imputation performance in the case of low-coverage European genomes, including ancient genomes, especially at rare variants and lower depths of coverage[75]. In the case of ancient DNA, when the target genome is not well represented by modern reference populations or when a boost in imputation accuracy is required, additional reference panels can be assembled with high-quality ancient genomes of individuals with more closely shared ancestry. Furthermore, the number of sequenced ancient genomes has been growing exponentially and with no sign of slowing down. This means that more and more ancient genomes will be available with different ancestries and from different time periods and with that comes the opportunity to expand existing reference panels with ancient genomes and to implement imputation in a more standardized way.

## Methods

In this section, we describe the methods implementation, starting with the Koszyce ancient trio data generation, followed by imputation, that includes all the file processing, imputation using GLIMPSE and using Beagle4.1, then the three downstream applications (PCA, genetic clustering analyses and ROH) and finishing with the reference data sets used in this study. All post-imputation analyses and corresponding plots were produced using python v3.6.12 and R v4.0.3.

### The Koszyce ancient trio data generation

The Koszyce ancient trio (mother, father and son) was originally sequenced in ref. [47] and re-sequenced to higher coverage in the context of this study. The DNA was extracted from petrous bone excavated from a Late Neolithic mass grave in Koszyce, in what is today Poland.

Using the same DNA extracts as ref. [47], two additional double-stranded libraries per sample were constructed based on ref. [76]. This was followed by enzymatic USER treatment to remove DNA damaged sites in the form of uracils. The optimal PCR cycle number was determined by qPCR. Indexed and amplified libraries were purified, quantified on an Agilent Bioanalyzer 2100, and pooled equimolarly. The

pooled libraries were then sequenced on two Novaseq lanes (150 paired-end reads).

The sequenced reads mapping was performed as in ref. [24]. The sequenced reads were aligned to both the human reference genome build 37 and the mitochondrial genome (rCRS). After alignment, reads were filtered based on a mapping quality threshold of 30 and sorted using Picard v.1.127 (http://broadinstitute.github.io/picard/) and samtools[51]. The resulting data was merged at the library level and duplicates were removed using Picard MarkDuplicates v.1.127. The merged data was then consolidated at the sample level. To improve the accuracy of the alignment, sample-level BAM files were realigned using GATK[77] v.3.3.0. Subsequently, the md-tag was updated and extended base alignment qualities (BAQs) were calculated using samtools calmd v.1.10.

## Estimating damage patterns
The frequency of C-to-T mismatches at the 5′ end of the aligned reads were estimated using bamdamage[73].

## Trimming the reads' ends in bam files
To test the effect of trimming the ends of the aligned reads on imputation accuracy (Supplementary Note 2), we used BamUtil[78] v1.0.14 to trim five base pairs from each end of the reads.

## Imputation
**File processing prior to imputation.** We downsampled high-coverage (10x-59x range) ancient genomes to coverages 0.1x, 0.25x, 0.5x, 0.75x, 1.0x, and 2.0x, using samtools[51] v1.10. The subsampling fraction was determined by first calculating the average coverage across the variant sites in the 1000 Genomes phase 3 reference panel[17] phased with TOPMed[18] (see Datasets section) so that the resulting downsampled genome had the intended coverage at those sites. Then, we computed genotype likelihoods for the downsampled and the original high-coverage genomes for the abovementioned variant sites.

To generate the genotype calls and genotype likelihoods, we used bcftools[51] v1.10 and, as default, the command bcftools mpileup with parameters -I -E -a "FORMAT/DP" --ignore-RG, followed by bcftools call -Aim -C alleles. To call genotypes from the high-coverage genomes, we have applied additional parameters for quality control (more details below).

We also generated both genotype calls from the high-coverage genomes and genotype likelihoods for the downsampled data (1x) with ATLAS[79] v0.9.9 (see Supplementary Note 1 and Supplementary Note 2) using the MLE caller and the empirical post-mortem damage pattern observed across reads, as described in https://bitbucket.org/wegmannlab/atlas/wiki. For sake of time, we skipped the first step, splitMerge, that separates single-end alignments by length and merges the mates of paired-end reads and requires specification of the different libraries contained in a bam file. It is often the case that an ancient genome is obtained from a mixture of paired-end and single-end libraries. We observed that this first step we skipped did not have much impact when the bam files only had single-end libraries, but the genotype calling was seemingly less accurate when there were paired-end libraries in the bam files. So, we do not report here results we obtained from ATLAS calls from ancient genomes that were sequenced from paired-end libraries.

To obtain a trimmed validation dataset (Supplementary Note 2), we trimmed five base pairs at both ends of the reads using the command trimBam from the package bamutil[78] v1.0.14. Then, we called genotypes using bcftools v1.10, as previously described.

The final validation dataset was obtained by implementing the following filtering approach:[38] i) genotype calling with bcftools v1.10 with mapping and base quality filters of 30 and 20 (-q 30 -Q 20), respectively, and with the parameter -C 50, as recommended by the SAMtools developers for BWA mapped data to reduce

mapping quality for reads with an excess of mismatches; ii) exclusion of the sites that are not in the 1000 Genomes accessible genome strict mask;[80] iii) removal of sites located in regions known to contain repeats (RepeatMask regions in UCSC Table Browser[81], http://genome.ucsc.edu/); iv) filtering out sites with extreme values of depth of coverage when comparing to the average genome coverage: below the maximum of one third of the mean depth of coverage (DoC) and eight, that is, $\max(\frac{DoC}{3},8)$, and depth above twice the average depth; v) filtering out of sites with the field QUAL below 30.

**Imputation using GLIMPSE.** We imputed the downsampled genomes using GLIMPSE[13] v1.1.1. First, we used GLIMPSE_chunk to split chromosomes into chunks of sizes in the range 1–2 Mb and included a 200-kb buffer region at each side of a chunk. Second, imputation was performed with GLIMPSE_phase on the chunks with parameters --burn 10, --main 15 and --pbwt-depth 2, with 1000 Genomes as the reference panel. And then, we ligated the imputed chunks with GLIMPSE_ligate.

**Imputation using Beagle4.1.** To evaluate how GLIMPSE performs compared to Beagle4.1[11] regarding imputation of low-coverage ancient genomes, we imputed the same data, but restricted to 1.0x, with Beagle4.1 with parameters --modelscale 2 and --niterations 0, that represent a trade-off between accurate results and running times.

**Imputation accuracy evaluation.** We used GLIMPSE_concordance to quantify imputation accuracy and genotype concordance, having the high-coverage data as validation. Only sites that were covered by at least eight reads and whose genotypes have a posterior probability of 0.9999 or more were used in validation. With GLIMPSE_concordance we obtained (i) imputation accuracy, that is, the squared correlation between dosage fields VCF/DS (DS varies between 0 and 2 that can be seen as a mean genotype value obtained from the genotype probabilities: $DS = \sum_{i=0}^{2} iGP_i$, where $GP_i$ is the genotype probability for genotype $i$) in imputed and validation datasets, divided in MAF bins, and (ii) genotype discordance, i.e., proportion of sites for which the most likely imputed genotype is different from the corresponding validation genotype for homozygous reference allele (RR), heterozygous (RA) and homozygous alternative allele sites (AA). We also estimated non-reference-discordance, NRD, defined as $NRD = (e_{RR} + e_{RA} + e_{AA})/(m_{RA} + m_{AA} + e_{RR} + e_{RA} + e_{AA})$, where $e_X$ and $m_X$ stand for the number of errors and matches at sites of type X, respectively. NRD is an error rate which excludes the number of correctly imputed homozygous reference allele sites, which are the majority, thus giving more weight to imputation errors at alternative allele sites.

## Testing significance of Spearman correlation between sample age and imputation accuracy
We calculated Spearman correlation using the function spearmanr from the python package scipy.stats. We performed a two-sided permutation test with 10,000 permutations to test whether the estimated correlation was significantly different from zero.

## Downstream analyses
**File processing.** We filtered the imputed data by imposing that, for each variant site, the genotype probability (VCF/GP) for the most confidently imputed genotype to be at least 0.80. Then, we generated two datasets with different minor allele frequency (MAF) filters: MAF > 5% (6,550,734 SNPs) for the data used in PCA and ROH analyses, and MAF > 1% (11,553,877 SNPs) for admixture analysis, since with stricter MAF filters we would lose sites that distinguish the different populations. We used PLINK[82] v1.90 to merge 1000 Genomes, high-coverage and imputed data into one file. In the case of PCA and admixture analyses, we intersected the resulting sites with the ones present in the Allen Ancient DNA Resource (AADR) data genotyped

at the 1240K array sites[63], that we refer to as the "1240K dataset" hereafter.

**PCA.** We performed PCA with smartpca (eigensoft[83] package v7.2.1) without outlier removal (*outliermode: 2*). The 10 first principal components (*numoutevec: 10*) were calculated using the 1000 Genomes genetic data and both the imputed and high-coverage data were projected onto the resulting components (*lsqproject: YES*).

To perform the t-tests to test if there were significant differences in coordinates between validation and corresponding 1x imputed genomes for the first 10 principal components, we used the default R function *t.test*, running it in unpaired mode to test whether the mean of the differences was significantly different from 0 with a two-sided alternative hypothesis.

**Admixture analysis.** We estimated admixture proportions for 21 ancient Europeans with the software ADMIXTURE[67] v1.3.0 in unsupervised mode. For the reference panel, we used a subset of the 1240K dataset containing nine western hunter gatherers, 26 Anatolian farmers and 26 individuals of Steppe ancestry[63] (see Supplementary Table 5). Contrary to the imputed and high-coverage genomes, the reference data are pseudo-haploid. We merged the reference panel with each of the imputation datasets (different coverages) with plink v1.90. We removed sites that were missing in more than 30% of the individuals. We proceeded similarly for the high-coverage dataset. We ran ADMIXTURE on seven configurations: merged reference panel and high-coverage individuals, and merged reference panel with each of the six imputed data sets (with initial coverage between 0.1x and 2.0x). For each configuration and number of clusters, we ran ADMIXTURE for K between two and five with 20 replicates (20 different seeds) and chose the replicate that yielded the largest log-likelihood value. In the final run, we obtained the standard error and bias of the admixture estimates using the option *--B 1000* that calculates these quantities with bootstrapping and 1000 replicates.

**Runs of homozygosity (ROH).** We estimated ROH with plink v1.90 with the parameters[72] *--homozyg, --homozyg-density* 50, *--homozyg-gap* 100, *--homozyg-kb* 500, *--homozyg-snp* 50, *--homozyg-window-het* 1, *--homozyg- window-snp* 50 and *--homozyg-window-threshold* 0.05. We estimated ROH twice: i) using transversion sites only, thus excluding sites that can be affected by aDNA damage, and ii) using both transversions and transitions.

**Datasets**
**Ancient genomes in this study.** The 43 downsampled and imputed ancient genomes (Supplementary Table 1) were obtained from the "Ancient Genomes dataset" that was compiled in the context of the study of ref.[24].

**Reference panel for imputation.** We used a version of 1000 Genomes v5 phase 3 (2504 genomes)[17], where the genomes were re-sequenced at 30x, and subsequently phased using TOPMed[18], and with sites present in TOPMed. These data are available in European Nucleotide Archive, under project PRJEB31736 and secondary study accession ERP114329. Only biallelic sites were retained (~90 million SNPs). This panel was lifted over from build 38 to hg19 reference genome assembly using Picard liftoverVCF v1.18.11 (https://gatk.broadinstitute.org/hc/en-us/articles/360037060932-LiftoverVcf-Picard-), with hg38ToHg19 chain from the University of California, Santa Cruz liftOver tool (http://hgdownload.cse.ucsc.edu/goldenpath/hg38/liftOver/).

**Present-day European genomes.** This dataset consists of a subset of 23 European genomes from the Simons Genome Diversity Project (SGDP)[84], as specified in Supplementary Table 3. We downloaded the corresponding bam files aligned to the hg19 reference genome

from the Seven Bridges Cancer Genomics Cloud (https://www.cancergenomicscloud.org). We downsampled the data to 1x and imputed as before.

**Reference panel for genetic clustering analyses.** The Allen Ancient DNA Resource (AADR)[85] that we refer to as "1240K dataset", is publicly available at https://reich.hms.harvard.edu/allen-ancient-dna-resource-aadr-downloadable-genotypes-present-day-and-ancient-dna-data.

We extracted a subset of the 1240K dataset[63] containing ancient individuals of the three ancestries we were interested in: 26 Anatolian farmers (Anatolia_N), 26 Steppe individuals (Steppe_EMBA), and nine western-hunter gatherers (WHG), as specified in Supplementary Table 5, to the exclusion of Loschbour, a genome that was also included in the dataset of 42 high-coverage genomes that we downsampled and imputed. We converted this subset from eigenstrat format to plink bed using the convertf command (eigensoft package v7.2.1). After that, we used plink v1.190 to do all of the data handling, such as merging plink bed files and filtering out sites with high missingness.

**Reporting summary**
Further information on research design is available in the Nature Portfolio Reporting Summary linked to this article.

## Data availability
All data supporting the findings described in this manuscript are available in the article and its Supplementary Information files, public repositories and from the corresponding author upon request. The Koszyce ancient trio data (RISE1159, RISE1160, RISE1168) generated in this study have been deposited in the European Nucleotide Archive (ENA) database under accession code PRJEB61632. The unfiltered imputed ancient genomes (original genomes were downsampled to depths of coverage in the range 0.1x–2.0x) are available in Zenodo (https://doi.org/10.5281/zenodo.7993392). The 1000 Genomes Project phase 3: 30X coverage whole genome sequencing data is available at the European Nucleotide Archive, under project PRJEB31736 and secondary study accession ERP114329 (https://www.ebi.ac.uk/ena/browser/view/PRJEB31736). The SGDP bam files aligned to hg19 reference genome were downloaded from Seven Bridges Cancer Genomics Cloud. The AADR[85] dataset is publicly available at https://reich.hms.harvard.edu/allen-ancient-dna-resource-aadr-downloadable-genotypes-present-day-and-ancient-dna-data. The remaining 40 ancient human genomes in this study have origin on the following studies: atp016[30] (https://doi.org/10.1073/pnas.1717762115); Stuttgart & Loschbour[41] (https://doi.org/10.1038/nature13673); Ballynahatty & Rathlin1[44] (https://doi.org/10.1073/pnas.1518445113); sf12[45] (https://doi.org/10.1371/journal.pbio.2003703); NE1 & BR2[46] (https://doi.org/10.1038/ncomms6257); SIII[48] (https://doi.org/10.1126/science.aao1807); SSG-A-2, HSJ-A-1 & STT-A-2[49] (https://doi.org/10.1126/science.aar2625); VK1[50] (https://doi.org/10.1038/s41586-020-2688-8); SZ15, SZ3, SZ4, SZ45, SZ43 & SZ1[31] (https://doi.org/10.1038/s41467-018-06024-4); baa01, ela01 & new01[32] (https://doi.org/10.1126/science.aao6266); I10871[33] (https://doi.org/10.1038/s41586-020-1929-1); Mota[34] (https://doi.org/10.1126/science.aad2879); KK1[35] (https://doi.org/10.1038/ncomms9912); WC1[36] (https://doi.org/10.1126/science.aaf7943); BOT2016 & Yamnaya[37] (https://doi.org/10.1126/science.aar7711); Andaman, AHUR_2064, Lovelock2, Lovelock3, Clovis, Sumidouro5, A460[38] (https://doi.org/10.1126/science.aav2621); USR1[42] (https://doi.org/10.1038/nature25173); Saqqaq[43] (https://doi.org/10.1038/nature08835); Ust'-Ishim[39] (https://doi.org/10.1038/nature13810); Kolyma_River & Yana[40] (https://doi.org/10.1038/s41586-019-1279-z).

## Code availability
The scripts we used to impute the ancient genomes, as well pre- and post-processing steps can be found in the following github repository:[86] https://github.com/bsmota/aDNA_imputation.

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

## Acknowledgements

We are thankful to Isabel Alves, Samuel Neuenschwander and J. Víctor Moreno-Mayar for fruitful discussions that contributed to improving this study. B.s.d.M. was supported by a Swiss National Science Foundation (SNSF) project grant (PP00P3_176977) to O.D. and by a European Research Council grant (grant agreement no. 679330) to A.-S.M. S.R. was supported by Swiss National Science Foundation (SNSF) project grant (PP00P3_176977). D.I.C.D. was supported by the European Research Council grant (grant agreement no. 679330) to A.-S.M. C.E.G.A. was supported by the National Institute of General Medical Sciences of the National Institutes of Health under award number R35GM142939. N.N.J. was supported by Aarhus University Research Foundation. H.S. was supported by the European Research Council (grant agreement no. 101045643).

## Author contributions

B.s.d.M., A.-S.M., and O.D. designed the study and drafted the paper. B.s.d.M. and O.D. performed the experiments. S.R. helped with imputation. D.I.C.D., C.E.G.A, M.E.A., M.S. and E.W. helped with the population genetics analyses. H.S., M.E.A., N.N.J., M.H.S., P.W., A.S., M.M.P. generated and provided the ancient trio data. This work has been

supervised by O.D. and A.-S.M. All authors helped with interpretation and reviewed the final manuscript.

## Competing interests

The authors declare no competing interests.
