## [Peer Review File · Nature Communications]

Imputation of ancient human genomesREVIEWER COMMENTS

Reviewer #1 (Remarks to the Author):

This work investigates imputing ancient DNA (aDNA) when using a published method that works well for low-coverage modern DNA data (called GLIMPSE). For this purpose, the authors utilized a dataset of 43 previously published high-coverage whole-genome-sequenced aDNA samples (mostly 10x or more). They compare the accuracy of imputation when downsampling this DNA data to lower coverage (0.1-2x) and comparing imputed genotypes to diploid genotype calls of the full data. Additionally, the authors investigate biases in downstream applications (i.e. PCA, Admixture, and ROH calling on imputed data), and test the lower coverage limit of these analyses. Finally, also the accuracy of phasing GLIMPSE data is tested, based on an ancient parent-offspring trio (Copper Age Europe, with high-quality aDNA reported elsewhere in another publication) that allows comparing such computational phasing to the "gold-standard" trio phasing. Notably, imputation and the tested downstream analysis work surprisingly well down to at least 0.5x, especially for common SNPs, for ancient individuals as old as 40k years, and for all tested individuals that share the out-of-Africa bottleneck.

While no new data or any new findings about the human past are presented, nor any new computational method is introduced, exploring the limits of imputing aDNA is an important technical step. Several published studies have already begun to use such imputed aDNA data - without much exploration of performance accompanying it, as imputation has the potential to unlock a wide range of downstream applications requiring diploid data. While aspects of the imputation of aDNA have been investigated before in published work, this study pioneers exploring a wide range of global samples (from four continents) and ages.

This work is therefore clearly of interest to the aDNA community. An important limitation of significance for the field though is that the tested data is all whole-genome sequencing data, which is only constituting a minority of the genome-wide aDNA record. Instead, in-solution capture on the so-called 1240k SNPs is a widely used technique (in the AADR Ressource v52.1, a comprehensive record of published aDNA work as of 2022, 6001 of all 8413 human aDNA samples are 1240k capture data!). This 1240k SNP capture data is genome-wide data, therefore imputation will likely perform well too for this use case, the question is only to what coverage limit. In principle, the same imputation pipeline as tested here could also test this key use case for human aDNA studies.

Generally, this article is technically competent, and there are no major flaws questioning the depicted results. I only have a few relatively minor comments on the interpretation of some results, and regarding checking the validation data below (see section A).

Beyond adding tests for 1240k data, I have some additional suggestions for improving the significance of the manuscript, please find them below as well (section B). I stress that none of those suggestions are make-or-break. Making such comments encouraging further work is very easy to state for a reviewer for such a technical testing article (there is always more to explore). So if the authors feel that these experiments are not a good use of their time or are outside the current scope of the article those comments should be skipped.

The article is well written, uses simple nice visualizations of high quality and I find it generally easy to follow. Relevant work seems to be cited well as far as I can see.

A) Major Comments

A1) Was deduplication applied to the ground truth data? Removing PCR duplicates, e.g. with "MarkDuplicates (Picard)", is important for over-sequenced libraries of limited complexity for which there can be a very substantial amount of PCR duplicate reads in aDNA data. And these duplicates can severely throw genotype calling off by making it look like there are more independent reads than there really are. In case this has been done it should be explicitly mentioned, and if not it would be very reassuring to do it.

A2) There is an important point to be made regarding the specific downstream analysis tested here. PCA & Admixture can and actually are routinely run on pseudo-haploid data - and works for a coverage lower than the "bias" threshold here (0.1x instead of 0.25-0.5x), and this is not shown. So why would someone use impute that data for that analysis? It is okay here to say that this analysis testing is done to test whether imputation introduces subtle pop-gen biases. But one should likely avoid giving the impression that PCA/Admixture on imputed data is endorsed over applying it to pseudo-haploid data - unless clearly specifying what are the benefits.

A3) I could not find a code availability statement: While only publicly available software was used as indicated in the reporting summary, this study applied novel custom bioinformatic pipelines and scripts. It would be very helpful to share these functions e.g. in a GitHub repository, not only for reproducibility, but especially if the imputation pipeline here should serve as a blueprint for other aDNA studies as likely intended.

A4) L147: "We found that imputation accuracy of ancient genomes was similar to the accuracy reported for present-day genomes when using the same imputation method". This is a key claim of the paper, highlighted prominently several times. However, the performance of imputation on modern data is only

referred to via a citation but never directly compared to. It would be very helpful to add a Supp. table or Supp. figure showing / quantifying that claim, or maybe even adding a (downsampled) modern genome into a main analysis or figure panel. Comparing performance across figures in two publications is hard on the reader and also leaves some room for biases due to different pipelines.

B) Suggested Work improving the impact & novelty (optional)

B1) There is a lingering finding that imputation is able to "remove" effects of aDNA damage (see e.g. ROH calling of Sumidouro5, where the validation set is obviously affected by the large amounts of aDNA damage but the imputed dataset seems to work very well at ~1x) - but more work would be needed to work that one out. An interesting finding could be to check whether imputation even of a high coverage sample can in fact help to correct erroneous diploid genotypes (such as C->T hets) or at least weaken the certainty - maybe even effectively outperforming ANGSD via using haplotype information.

B2) The effects of global ancestry on imputation accuracy are very nicely worked out in the main figures, but one very salient pattern seems to be connected to sample age (for Out-of-Africa ancestry samples). The three by far the oldest samples (>30ky), Ust Ishim, Yana, & SIII, all have exceptionally low imputation performance for rare variants (Fig. 2b). That seems definitely a pattern worth exploring (or at least mentioning), as it gives insights on the temporal limits of imputation (the bigger the temporal distance to the ref panel, the harder the imputation as shared haplotypes used by the imputation HMM will be shorter). Generally, this correlation could be explored, as there is theoretical reason to believe that older samples generally work less well for imputation than modern samples (see e.g. Biddanda et al. 2021, <https://doi.org/10.1093/genetics/iyac038> for a theoretical treatment of that simple intuition).

B3) - Evaluating the interpretability of imputed genotype probabilities (GPs). The manuscript focuses on looking into how GP threshold filtering affects the accuracy of imputed data and the number of filtered sites. But for downstream applications, it would be extremely useful to know whether these GPs actually correspond even on an order of magnitude to the real uncertainties of genotypes. Many downstream population genetic methods can take such uncertainties into account and can model them. You have an ideal test case to compare to the actual genotype here. Even if these GPs should not be taken at face value, it would be useful to describe this pattern.

C) Minor Comments

C1) Tab. S1: The coverage column is rounded to integers - but giving at least one post-comma digit seems in order for samples with coverages $\sim 10x$. It would perhaps help to additionally add it as a computer-readable table (e.g. a proper supplementary table), and maybe also add more highly relevant information (such as UDG treatment, and estimated damage rates on terminal reads).

C2) How is coverage calculated? Is it coverage on the mappable part of the genome, or the number of all sequences divided by the total genome length? It would help to specify the key quantity.

C3) The authors use a minimum lower cutoff of $8x$ for diploid genotype calling per site. This cutoff raises some questions for the $5.4x$ ground truth sample prominently featured in the trio (RISE1160): Was there any special attention for this sample as with $8x$ filtering one would filter out most sites? Was the test set of validation SNPs for the trio a subset? There is even the possibility to do trio-aware genotype calling, to enforce Mendelian compatibility (that could help with a $5x$ genome which otherwise is very hard to genotype).

C3) Fig. 2: If find the behavior of the NRD non-concordance rate slightly strange, i.e. that it can be higher than all three RR/RA/AA concordance rates. Is this a standard measure? It is likely because RR errors are counted in the numerator, but RR matches not in the denominator - leading to it not really being interpretable as a rate.

C4) L34: "We imputed most of the 42 high-coverage genomes downsampled to $1x$ with low error rates (below 5%)" You could give the number here, "most" is a bit vague.

C5) L60: "Mostly, the Li and Stephens model of linkage disequilibrium (LD) is at the core of this HMM". I am being very anal now, but it is actually haplotype info instead of LD: Even if there is no LD in a population Li&Stephens HMM imputation would work if there are e.g. sporadic shared haplotypes between distant relatives (IBD segments). Of course, the original article frames it in terms of LD too...

C6) L134: "with about half of the individuals being from Europe and the other half from Africa, America, and Asia". Explicitly stating the numbers could save space and be more accurate.

C7) Giving switch error rates per cM would help (or the average cM of haplotype phased correctly), as they are more general and could even be compared across different SNP sets. It's also a property important for haplotype copying HMMs (see Biddanda et al. 2021, they looked into the expected switch rate in the Li and Stephens haplotype copying models in ancient samples).

C8) - L184: "This suggests that having haplotypes in the reference panel that match the ancestry of the target haplotypes is fundamental to achieve high imputation accuracy, even if these reference haplotypes originate from admixed individuals."

Seeing that also 45ky old Ust Ishim is imputed fairly well (more than twice as old as the Native American split from Eurasians) - makes me believe it could actually be enough that the reference panel is simply not too far away in terms of coalescent distance from the target for imputation to work well. It could well be that the limit is the out-of-Africa bottleneck, and e.g. Native Americans can be imputed well with modern Eurasian haplotypes, at least for common variants.

To substantiate your alternative claims that it is in fact the admixed modern Native Americans in the reference panel that help imputation for ancient Native Americans you could try removing those from the reference panel - according to your hypothesis imputation quality should break down drastically then.

Reviewer #2 (Remarks to the Author):

The authors provide exhaustive simulation results for imputation of ancient genomes, considering genotype accuracy, as well as reproduction of runs of homozygosity and PCA with imputed results.

This is accomplished using a relatively diverse panel of individuals, avoiding the rather common bias towards individuals of European descent and European geographical origin. The conclusions also highlight the challenges for African populations. On a positive side, good results are achieved for American populations, despite the fact that the reference panels used only included modern individuals post-contact with a decent amount of admixture with other descents.

Overall, including in the abstract, I am surprised at the strong focus on degradation, rather than contamination. Contamination with bacterial and to some extent fungal DNA is, in my experience, a main contributor to low effective coverage for human DNA (while contamination with other human DNA can be a much larger problems for samples). Contamination is discussed briefly, but only late in the manuscript. This does not affect the conclusions, but it did strike me as odd in the description of the state of the art.

The methodology is sound and the comparison against Glimpse is illuminating. The manuscript does a good job of relating existing work on aDNA imputation in a factual sense, but I think it uses value words,

especially in the abstract, but also in the introduction and discussion, that to some extent overstate the novelty of this work.

The writing is a bit wordy and informal at times. I think some shortenings and general polish work would improve the text.

In the results section, phrases "no indication that imputation introduced any substantial bias" line 317, "we did not observe substantial differences" line 155, "close to the validation ROH" line 332 sometimes seem somewhat debatable. Interpreting the underlying figures and data is somewhat subjective and I am not totally convinced that I would agree in all cases. Some attempts are made to justify these claims by statistical tests or metrics of differences, but not a lot. I see this as a weakness.

I think the justification for the various MAF thresholds (1% vs 5%) is not very clear.

A minor comment is also that at least some references seem to lack volume information completely, despite them referring to journals using a volume numbering.

REVIEWER COMMENTS

Reviewer #1 (Remarks to the Author):

This work investigates imputing ancient DNA (aDNA) when using a published method that works well for low-coverage modern DNA data (called GLIMPSE). For this purpose, the authors utilized a dataset of 43 previously published high-coverage whole-genome-sequenced aDNA samples (mostly 10x or more). They compare the accuracy of imputation when downsampling this DNA data to lower coverage (0.1-2x) and comparing imputed genotypes to diploid genotype calls of the full data. Additionally, the authors investigate biases in downstream applications (i.e. PCA, Admixture, and ROH calling on imputed data), and test the lower coverage limit of these analyses. Finally, also the accuracy of phasing GLIMPSE data is tested, based on an ancient parent-offspring trio (Copper Age Europe, with high-quality aDNA reported elsewhere in another publication) that allows comparing such computational phasing to the "gold-standard" trio phasing. Notably, imputation and the tested downstream analysis work surprisingly well down to at least 0.5x, especially for common SNPs, for ancient individuals as old as 40k years, and for all tested individuals that share the out-of-Africa bottleneck.

While no new data or any new findings about the human past are presented, nor any new computational method is introduced, exploring the limits of imputing aDNA is an important technical step. Several published studies have already begun to use such imputed aDNA data - without much exploration of performance accompanying it, as imputation has the potential to unlock a wide range of downstream applications requiring diploid data. While aspects of the imputation of aDNA have been investigated before in published work, this study pioneers exploring a wide range of global samples (from four continents) and ages.

This work is therefore clearly of interest to the aDNA community. An important limitation of significance for the field though is that the tested data is all whole-genome sequencing data, which is only constituting a minority of the genome-wide aDNA record. Instead, in-solution capture on the so-called 1240k SNPs is a widely used technique (in the AADR Ressource v52.1, a comprehensive record of published aDNA work as of 2022, 6001 of all 8413 human aDNA samples are 1240k capture data!). This 1240k SNP capture data is genome-wide data, therefore imputation will likely perform well too for this use case, the question is only to what coverage limit. In principle, the same imputation pipeline as tested here could also test this key use case for human aDNA studies.

Generally, this article is technically competent, and there are no major flaws questioning the depicted results. I only have a few relatively minor comments on the interpretation of some results, and regarding checking the validation data below (see section A).

Beyond adding tests for 1240k data, I have some additional suggestions for improving the significance of the manuscript, please find them below as well (section B). I stress that none of those suggestions are make-or-break. Making such comments encouraging further work is very easy to state for a reviewer for such a technical testing article (there is always more to explore). So if the authors feel that these experiments are not a good use of their time or are outside the current scope of the article those comments should be skipped.

The article is well written, uses simple nice visualizations of high quality and I find it generally easy to follow. Relevant work seems to be cited well as far as I can see.

We thank the reviewer for the thorough summary and comments. Regarding assessing imputation accuracy in the case of data sequencing using in-solution capture, we agree that it is of high interest for the ancient DNA community. We therefore imputed five capture-

sequenced genomes for which we also have high-coverage shotgun-sequenced data. We describe the analysis at the end of the results section (“1. Accuracy of low-coverage ancient DNA imputation”) with the title “Imputation of ancient capture data: increased accuracy at the capture sites”. The analysis results can be found in Figure 5 and in the supplementary material file (Figure S8 and Figure S9).

A) Major Comments

A1) Was deduplication applied to the ground truth data? Removing PCR duplicates, e.g. with "MarkDuplicates (Picard)", is important for over-sequenced libraries of limited complexity for which there can be a very substantial amount of PCR duplicate reads in aDNA data. And these duplicates can severely throw genotype calling off by making it look like there are more independent reads than there really are. In case this has been done it should be explicitly mentioned, and if not it would be very reassuring to do it.

We agree that keeping PCR duplicates would impact genotype calling, particularly the validation data. We carefully verified that all publications for the 42 bam files report to have carried out this step.

Nevertheless, we ran “MarkDuplicates” from picard-tools on the 42 high-coverage bam files. We found that 11 of them had higher than expected duplication rates, between 5.7% and 17.6%. To verify if removing these newly marked duplicates produced different genotypes, we ran bcftools stats on both these newly called genotypes and the original validation data, and we found NRD values below 0.5% (see Table R1, below). We concluded that this duplicate removal has negligible impact on the validation data, which (i) supports that this step had already been performed and (ii) implies that results will not change if we re-apply this filtering.

It is not clear why we detect duplicates when these were reportedly removed in their original studies. There are multiple possible explanations for this, including the usage of different duplicate removal methods with different levels of stringency. Another possibility is the renaming and/or merging of the different libraries in such a way that these can no longer be distinguished. For nine of these genomes, only one library was specified in the bam file header, even though sequencing ancient genomes at high coverage requires multiple libraries. This means that “MarkDuplicates” compares reads from different libraries between each other and identifies seemingly duplicated reads, which are not PCR duplicates by definition (because they come from different libraries).

Table R1: Individual samples for which we found non-negligible duplication rates. We report the number of libraries in the output of “MarkDuplicates”, the mean duplication rate across the different libraries and non-reference discordance (NRD) between the genotypes before and after re-applying “MarkDuplicates”.

ID	Number libraries in output	Duplication rate (%)	NRD (%)
atp016	1	14.9	0.477
baa01	1	17.6	0.290
BOT2016	1	6.0	0.153
ela01	1	15.6	0.101
I10871	4	16.4	0.080
new01	1	16.5	0.361
sf12	1	37.1	0.019
UstIshim	9	5.7	0.023
WC1	1	4.8	0.151
Yamnaya	1	14.5	0.207
Loschbour	1	14.5	0.065

A2) There is an important point to be made regarding the specific downstream analysis tested here. PCA & Admixture can and actually are routinely run on pseudo-haploid data - and works for a coverage lower than the "bias" threshold here (0.1x instead of 0.25-0.5x), and this is not shown. So why would someone use impute that data for that analysis? It is okay here to say that this analysis testing is done to test whether imputation introduces subtle pop-gen biases. But one should likely avoid giving the impression that PCA/Admixture on imputed data is endorsed over applying it to pseudo-haploid data - unless clearly specifying what are the benefits.

We thank the reviewer for the comment and we clarified the role of PCA and admixture in our study. We mention that we do these two analyses to investigate biases introduced by imputation at the beginning of the second part of the Results section:

“In order to detect and quantify potential bias introduced by imputation, we compared the results of downstream analyses, namely, principal component analysis (PCA) and genetic clustering analyses, [...]”

We also added a paragraph to the discussion regarding the goal of our study:

“In aDNA studies, pseudo-haploid data generation is standard procedure to handle low-coverage genomes. Ancestry information can be recovered from analyses of these data, such as PCA and genetic clustering analyses which work well at coverages as low as 0.1x. While imputing genotypes has evident advantages over making them pseudo-haploid, the goal of this work was to assess imputation accuracy and potential biases introduced in downstream analyses and not to compare the two approaches.”

A3) I could not find a code availability statement: While only publicly available software was used as indicated in the reporting summary, this study applied novel custom bioinformatic

pipelines and scripts. It would be very helpful to share these functions e.g. in a GitHub repository, not only for reproducibility, but especially if the imputation pipeline here should serve as a blueprint for other aDNA studies as likely intended.

We added a “code availability” statement to the manuscript as follows:

“Code availability

The scripts we used to impute the ancient genomes, as well pre- and post-processing steps can be found in the following github repository

https://github.com/bsmota/aDNA_imputation/blob/main/README.md.”

A4) L147: "We found that imputation accuracy of ancient genomes was similar to the accuracy reported for present-day genomes when using the same imputation method". This is a key claim of the paper, highlighted prominently several times. However, the performance of imputation on modern data is only referred to via a citation but never directly compared to. It would be very helpful to add a Supp. table or Supp. figure showing / quantifying that claim, or maybe even adding a (downsampled) modern genome into a main analysis or figure panel. Comparing performance across figures in two publications is hard on the reader and also leaves some room for biases due to different pipelines.

We added a comparison between imputation performance of present-day and ancient genomes in supplementary section 4 to back up our claims (Figure S3).

B) Suggested Work improving the impact & novelty (optional)

B1) There is a lingering finding that imputation is able to "remove" effects of aDNA damage (see e.g. ROH calling of Sumidouro5, where the validation set is obviously affected by the large amounts of aDNA damage but the imputed dataset seems to work very well at ~1x) - but more work would be needed to work that one out. An interesting finding could be to check whether imputation even of a high coverage sample can in fact help to correct erroneous diploid genotypes (such as C->T hets) or at least weaken the certainty - maybe even effectively outperforming ANGSD via using haplotype information.

We appreciate the reviewer's suggestion and are glad to share the same perspective. We are currently modifying the GLIMPSE software to include a specific mode that will assist researchers in identifying sequencing errors in ancient DNA data. The aim is to locate genotype calls (from high or low coverage) that contradict the underlying linkage disequilibrium structure. This is based on similar techniques proposed in the early days of SNP arrays. However, we believe that this aspect of our work is beyond the scope of the current paper, as it involves a substantial amount of work that should be described and published independently.

B2) The effects of global ancestry on imputation accuracy are very nicely worked out in the main figures, but one very salient pattern seems to be connected to sample age (for Out-of-Africa ancestry samples). The three by far the oldest samples (>30ky), Ust Ishim, Yana, & SIII, all have exceptionally low imputation performance for rare variants (Fig. 2b). That seems definitely a pattern worth exploring (or at least mentioning), as it gives insights on the temporal limits of imputation (the bigger the temporal distance to the ref panel, the harder the imputation as shared haplotypes used by the imputation HMM will be shorter). Generally, this correlation could be explored, as there is theoretical reason to believe that older samples generally work less well for imputation than modern samples (see e.g. Bliddanda et al. 2021, <https://doi.org/10.1093/genetics/iyac038> for a theoretical treatment of that simple intuition).

This is a good suggestion and we added an analysis on the relationship between imputation accuracy and age per MAF bin. We report this analysis' results in the manuscript under the subheading "Imputation error rates below 5% for most non-African 1x genomes" and in Supplementary Section 8. To better make sense of these results, we estimated the pairwise allelic differences between the ancient genomes and the genomes in the reference panel, used here as a proxy for coalescent distance. We illustrate these differences in a new version of Figure 2.

B3) - Evaluating the interpretability of imputed genotype probabilities (GPs). The manuscript focuses on looking into how GP threshold filtering affects the accuracy of imputed data and the number of filtered sites. But for downstream applications, it would be extremely useful to know whether these GPs actually correspond even on an order of magnitude to the real uncertainties of genotypes. Many downstream population genetic methods can take such uncertainties into account and can model them. You have an ideal test case to compare to the actual genotype here. Even if these GPs should not be taken at face value, it would be useful to describe this pattern.

We thank the reviewer for the recommendation and we acknowledge the importance of interpreting posterior probabilities of imputation (GPs). Such calibration has been done in the GLIMPSE publication (Rubinacci et al., 2021), in the Extended Data Figure 6 (<https://www.nature.com/articles/s41588-020-00756-0/figures/10>). This shows that the model is relatively well calibrated, i.e., a GP of 0.80 corresponds to a concordance of around 0.80.

C) Minor Comments

C1) Tab. S1: The coverage column is rounded to integers - but giving at least one post-comma digit seems in order for samples with coverages ~10x. It would perhaps help to additionally add it as a computer-readable table (e.g. a proper supplementary table), and maybe also add more highly relevant information (such as UDG treatment, and estimated damage rates on terminal reads).

We now report coverage with one post-comma digit. Besides, we estimated C-to-T mismatch frequency at the 5' end of the reads, but we had some technical problems for some of them. For most of these cases, we list either the C-to-T frequencies that are reported in the publications or whether the sequenced libraries were UDG-treated (as according to the publications). For three of these genomes (SSG-A-2, HSJ-A-1, STT-A-2), we could not extract the C-to-T frequency from the publication (Ebenesersdottir et al., *Science* (2018)), since it was not possible to clearly distinguish between the different sample values in the plot.

C2) How is coverage calculated? Is it coverage on the mappable part of the genome, or the number of all sequences divided by the total genome length? It would help to specify the key quantity.

We listed the genome coverages as reported in the original publications. To make the definition uniform across the 43 genomes, we now also report the mean depth of coverage across the 1000 Genomes (umich) biallelic sites.

C3) The authors use a minimum lower cutoff of 8x for diploid genotype calling per site. This cutoff raises some questions for the 5.4x ground truth sample prominently featured in the trio (RISE1160): Was there any special attention for this sample as with 8x filtering one would filter out most sites? Was the test set of validation SNPs for the trio a subset? There is even the possibility to do trio-aware genotype calling, to enforce Mendelian compatibility (that could help with a 5x genome which otherwise is very hard to genotype).

We thank the reviewer for the question. In this case, we did not use the original genomes as validation. All three genomes were downsampled and imputed, but the imputation and phasing assessment were solely based on Mendel's rules of inheritance. To estimate imputation errors, we focused on sites that were not REF/REF across the three individuals and counted as errors the cases when parental and offspring imputed genotypes were not compatible according to Mendel's rules of inheritance. The switch errors were also estimated using the phased imputed data, as explained in Figure 3 caption: "A switch error is counted between two consecutive heterozygous genotypes when the reported haplotypes are not consistent with those derived from the trio."

C3) Fig. 2: If find the behavior of the NRD non-concordance rate slightly strange, i.e. that it can be higher than all three RR/RA/AA concordance rates. Is this a standard measure? It is likely because RR errors are counted in the numerator, but RR matches not in the denominator - leading to it not really being interpretable as a rate.

We agree with the reviewer that NRD is not the easiest measure to interpret. However, it has some advantages:

1. Single accuracy metric that focuses on the most difficult genotypes to impute,
2. Commonly used in multiple other studies (e.g., Arthur et al., *Bioinformatics* (2016); Wang et al., *GENETICS* (2020); Lencz et al., *Human genetics* (2018)),
3. Computed with an independent tool and does not rely on in-house code.

C4) L34: "We imputed most of the 42 high-coverage genomes downsampled to 1x with low error rates (below 5%)" You could give the number here, "most" is a bit vague.

We agree with the reviewer and changed the sentence:

"We imputed 36 of the 42 high-coverage genomes downsampled to 1x with low error rates (below 5%) [...]"

C5) L60: "Mostly, the Li and Stephen model of linkage disequilibrium (LD) is at the core of this HMM". I am being very anal now, but it is actually haplotype info instead of LD: Even if there is no LD in a population Li&Stephens HMM imputation would work if there are e.g. sporadic shared haplotypes between distant relatives (IBD segments). Of course, the original article frames it in terms of LD too...

We edited the text as follows:

"Mostly, the Li and Stephen model of linkage disequilibrium (LD) and haplotype sharing is at the core of this HMM,"

C6) L134: "with about half of the individuals being from Europe and the other half from Africa, America, and Asia". Explicitly stating the numbers could save space and be more accurate.

We changed the text accordingly:

"[...] as well as epoch and continent the ancient individuals lived in, with 22 individuals from Europe, five from Africa, eight from Asia and eight from the Americas (**Figure 1B**)."

C7) Giving switch error rates per cM would help (or the average cM of haplotype phased correctly), as they are more general and could even be compared across different SNP sets. It's also a property important for haplotype copying HMMs (see Biddanda et al. 2021, they looked into the expected switch rate in the Li and Stephens haplotype copying models in ancient samples).

We disagree with the reviewer's assertion that the number of switches per cM is unaffected by the SNP set. This metric is influenced by the minor allele frequency, as a higher SNP density leads to an increase in rare variants, which are known to pose difficulties in phasing and are more prone to causing disruptions in runs of correctly phased heterozygous genotypes. In our study, we focussed on the switch error rate, as it is the most widely used metric in the literature to evaluate haplotype quality.

C8) - L184: "This suggests that having haplotypes in the reference panel that match the ancestry of the target haplotypes is fundamental to achieve high imputation accuracy, even if these reference haplotypes originate from admixed individuals."

Seeing that also 45ky old Ust Ishim is imputed fairly well (more than twice as old as the Native American split from Eurasians) - makes me believe it could actually be enough that the reference panel is simply not too far away in terms of coalescent distance from the target for imputation to work well. It could well be that the limit is the out-of-Africa bottleneck, and e.g. Native Americans can be imputed well with modern Eurasian haplotypes, at least for common variants.

To substantiate your alternative claims that it is in fact the admixed modern Native Americans in the reference panel that help imputation for ancient Native Americans you could try removing those from the reference panel - according to your hypothesis imputation quality should break down drastically then.

We thank the reviewer for the insightful comment. We agree that the suggested analysis substantiates the discussion. We then proceeded to subset the reference panel by removing one continental group at a time to keep the reference panel constant. We indeed observed a drop in imputation performance for the Native American ancient genomes when no American populations were in the subset reference panel. The drop was rather large at rare variants (MAF<5%), but indeed, as the reviewer mentioned, imputation accuracy was still high at common variants. The full description and results of this analysis can be found in Supplementary Section 7 and we mention it in the main text, under the results' subheading "Imputation error rates below 5% for most non-African 1x genomes", that we quote here:

"And yet, Native American genomes were also accurately imputed, even though the populations in the reference panel show different admixture moieties, ranging from low (e.g., Puerto Rican (PUR)) to high Native American (e.g., Peruvian (PEL)) admixture proportions. In fact, Figure 2B shows that the South American reference individuals tend to be genetically close to the Native American genomes (small pairwise allelic differences). We further confirmed the contribution of American reference haplotypes to the imputation of ancient Native American genomes by removing one continental group at a time from the reference panel. We found that imputation performance was only affected when using a reference panel without the American populations (see Supplementary Section 7). Imputation accuracy dropped to 0.46 from 0.78 at variants in the lowest MAF bin (0.1%-1.0%), while it was only slightly smaller (~0.97 vs. ~0.98) at common variants (MAF>5%), as shown in Figure S6."

Reviewer #2 (Remarks to the Author):

The authors provide exhaustive simulation results for imputation of ancient genomes, considering genotype accuracy, as well as reproduction of runs of homozygosity and PCA with imputed results.

This is accomplished using a relatively diverse panel of individuals, avoiding the rather common bias towards individuals of European descent and European geographical origin. The conclusions also highlight the challenges for African populations. On a positive side, good results are achieved for American populations, despite the fact that the reference panels used only included modern individuals post-contact with a decent amount of admixture with other descents.

Overall, including in the abstract, I am surprised at the strong focus on degradation, rather than contamination. Contamination with bacterial and to some extent fungal DNA is, in my experience, a main contributor to low effective coverage for human DNA (while contamination with other human DNA can be a much larger problem for samples). Contamination is discussed briefly, but only late in the manuscript. This does not affect the conclusions, but it did strike me as odd in the description of the state of the art.

We thank the reviewer for the comment. Contamination is indeed a major issue in ancient DNA studies. Therefore, we now mention it in the abstract:
“Due to postmortem DNA degradation and microbial colonization of remains, most ancient genomes sequenced to date have low depth of coverage”

And in the introduction:

“Ancient DNA (aDNA) is characterized by pervasive postmortem damage, including fragmentation and deamination. Moreover, the ubiquitous microbial contamination gives rise to, in most cases, low amounts of endogenous DNA, whereas contamination with DNA from related species is an even bigger challenge, as the original and the contaminant DNA cannot be easily deconvolved, and thus highly contaminated genomes are often discarded from the analyses.”

The methodology is sound and the comparison against Glimpse is illuminating.
We thank the reviewer for the positive summary.

The manuscript does a good job of relating existing work on aDNA imputation in a factual sense, but I think it uses value words, especially in the abstract, but also in the introduction and discussion, that to some extent overstate the novelty of this work. The writing is a bit wordy and informal at times. I think some shortenings and general polish work would improve the text. In the results section, phrases "no indication that imputation introduced any substantial bias" line 317, "we did not observe substantial differences" line 155, "close to the validation ROH" line 332 sometimes seem somewhat debatable. Interpreting the underlying figures and data is somewhat subjective and I am not totally convinced that I would agree in all cases. Some attempts are made to justify these claims by statistical tests or metrics of differences, but not a lot. I see this as a weakness.

We thank the reviewer for the comment. We changed the indicated sentences so as to make them more precise:

"we did not observe substantial differences" -> “[...] there were small differences in accuracy between imputed transversion and transition sites at rare variants ($0.1\% < \text{MAF} \leq 1\%$, $r^2=0.75$ and $r^2=0.77$ for transitions and transversions, respectively), but these differences disappeared for more common variants (Figure S4).”

"no indication that imputation introduced any substantial bias" -> "[...] that imputation introduced limited bias [...]"

"close to the validation ROH" -> "[...] the imputed genomes were close to the validation ROH, particularly for A460 (5% difference) and Ust'Ishim (0.7% difference)."

I think the justification for the various MAF thresholds (1% vs 5%) is not very clear. To conduct the different downstream analyses we did, we need variability in the data while controlling for the quality of the imputed sites. At first, we chose to remove SNPs with MAF below 5% and that is the threshold that we used for PCA and ROH analyses. However, with such a threshold, the genetic clustering analyses did not split the three populations (Steppe, WHG and Anatolian Farmers) at $K=3$. Only when we kept more rare variants by removing instead at $MAF < 1\%$, these three populations could be distinguished. And that is why we used a different MAF threshold for this analysis.

A minor comment is also that at least some references seem to lack volume information completely, despite them referring to journals using a volume numbering. We thank the reviewer for noticing that. We added volume information where it was missing.

REVIEWERS' COMMENTS

Reviewer #1 (Remarks to the Author):

The authors did a thorough and competent job in their revisions, and all my comments have been either directly addressed or clarified.

A series of additional experiments have investigated several exciting aspects of the findings. The work now makes clear that for ancient genomes of ancestry close to the reference panel, e.g. in Western Eurasia of the last few millennia, imputation works as well as for present-day genomes (e.g. Fig. S3 is striking!). If there is genealogical separation by more than a few thousand years, imputation for rare variants breaks down, while common variants can still be imputed even when separations are on the order of several 10ky (as shown by the imputation results for pre-European contact genomes from America or the imputation quality time series). These are key findings, helping to better establish the benefits and limits of imputation, not only of ancient DNA but in general.

I believe that this article is ready to publish as is and that it will become a primary reference for the key topic of imputing ancient DNA data.

My remaining suggestions are either typos or minor suggestions. The latter should only be addressed if the authors find them helpful. Please find them below:

#####

Minor Comments (A)

A1)

L473: "We, therefore, recommend a minimum depth of coverage of 2x at capture sites to attain accurate imputed calls, in the case of well-represented ancestries."

The additional experiments have expanded the scope of the work, now with important results for the imputation of data obtained with a standard capture approach widely used in aDNA (1240k SNPs). The authors tested downsampled coverage up 2x average depth, and from 1x to 2x observed still a big jump in imputation quality. That raises the question of whether the recommended 2x is where imputation

performance "plateaus", and whether 3x or 4x could help even further (as the full data of Stuttgart of 11.1x suggest a higher plateau, Fig. S8).

This recommended coverage is an important question that directly impacts future study designs, as capture data for well-preserved samples can often cost-effectively be extended to reach such coverages by simply adding additional sequencing depth.

I would therefore suggest extending the imputation tests to 3x-5x for the Stuttgart capture data, as this seems to be directly possible with little effort within the current testing framework.

A2)

Original Reviewer Comment:

C7) Giving switch error rates per cM would help (or the average cM of haplotype phased correctly), as they are more general and could even be compared across different SNP sets.

It's also a property important for haplotype copying HMMs (see Biddanda et al. 2021, they looked into the expected switch rate in the Li and Stephens haplotype copying models in ancient samples).

Author's Response:

We disagree with the reviewer's assertion that the number of switches per cM is unaffected by the SNP set. This metric is influenced by the minor allele frequency, as a higher SNP

density leads to an increase in rare variants, which are known to pose difficulties in phasing and are more prone to causing disruptions in runs of correctly phased heterozygous genotypes. In our study, we focussed on the switch error rate, as it is the most widely used metric in the literature to evaluate haplotype quality.

New Comment:

I did not want to assert that the number of switches per cM is unaffected by the SNP set - but merely that using switches per cM will allow comparing phasing quality in a metric that is key for downstream population genetic applications. E.g., the effects of different SNP sets mentioned in their rebuttal would become evident when comparing switch rates per cM (as this factors out the differences of SNP number).

Of course, the choice of metric is the authors and it is a valid argument to stay consistent with published literature.

However, I believe that it would be important to at least mention in a side sentence what the average genomic map distance is that can be phased correctly via imputation (on the 1000G SNP set) - as it has huge implications for downstream analysis such as LAI and ARG reconstruction whether average correctly phased segments are around 0.1, 1 or 10cM, say.

Computational imputation is the only viable way to obtain phased data for the vast majority of aDNA, and current downstream method developers often close their eyes and assume that computational phasing of aDNA is perfect and one obtains perfect long-range haplotypes (e.g. <https://doi.org/10.1101/2023.03.06.529121>).

This work could easily clarify the limits of the imputation-based phasing of ancient samples in terms of centimorgan so that for all population genetic method developers it becomes directly evident that assuming phasing on the order of several centimorgans is not viable for typical aDNA samples unless one uses trio-phasing. Talking about switch error rate per SNP number has the effect of hiding that from population genetic researchers.

A3) There has been some recent discussion on the blurry association between geography, ancestry, and genetic study group labels (e.g. <https://doi.org/10.48550/arXiv.2207.11595>).

Due to the nature of this work, this is a relevant topic here - for instance, the authors sometimes refer to "African genomes". I believe that here it is clear from the context, e.g. that the authors directly refer to geographic areas, relevant core genetic groupings (e.g. out-of-Africa), or sample labels from original studies (such as 1000G labels). But there is a multitude of opinions out there, and now that the article is close to being published, the authors might want to consider such issues just in case they have not already.

A4) The code availability statement links to the README file of the repository and not the master folder - is that intended?

Overall, these scripts are a very helpful addition - and I find it especially useful that also the resource settings of the various steps are shown as part of the cluster submission scripts.

A5) L61: "pseudo haploid" but usually "pseudo-haploid" throughout the manuscript

A6)

"Three out of the 43 ancient genomes in this study constitute a trio (mother, father, and son) that was recently re-sequenced and is not yet fully public (24,47)"

As there are co-authorships for this trio data, maybe the authors could consider releasing the three genomes also as part of this article or providing some transparent "available upon request" path (if the other article is not ready in time)? Trio-phasing will also be a crucial test for competing imputation methods, it would be fair to follow the aDNA standard of sharing publicly accessible data.

A7)

L470-472: "Furthermore, using five genomes that were both obtained via in-solution capture and shotgun sequencing (>10x for the latter), we found that imputation performance of capture-sequenced data was highest at the capture sites for rare variants."

I think you mean "highest at the capture sites for common variants"? Or do you refer to the fact that rare variants "on" the capture panel were imputed better than rare variants outside of it?

Reviewer #2 (Remarks to the Author):

I appreciate the authors largely addressing my previous comments. I am very happy with the current state of the manuscript.

Final reviewer comments

Reviewer #1 (Remarks to the Author):

The authors did a thorough and competent job in their revisions, and all my comments have been either directly addressed or clarified.

A series of additional experiments have investigated several exciting aspects of the findings. The work now makes clear that for ancient genomes of ancestry close to the reference panel, e.g. in Western Eurasia of the last few millennia, imputation works as well as for present-day genomes (e.g. Fig. S3 is striking!). If there is genealogical separation by more than a few thousand years, imputation for rare variants breaks down, while common variants can still be imputed even when separations are on the order of several 10ky (as shown by the imputation results for pre-European contact genomes from America or the imputation quality time series). These are key findings, helping to better establish the benefits and limits of imputation, not only of ancient DNA but in general.

I believe that this article is ready to publish as is and that it will become a primary reference for the key topic of imputing ancient DNA data.

My remaining suggestions are either typos or minor suggestions. The latter should only be addressed if the authors find them helpful. Please find them below:

#####

Minor Comments (A)

A1)

L473: "We, therefore, recommend a minimum depth of coverage of 2x at capture sites to attain accurate imputed calls, in the case of well-represented ancestries."

The additional experiments have expanded the scope of the work, now with important results for the imputation of data obtained with a standard capture approach widely used in aDNA (1240k SNPs). The authors tested downsampled coverage up 2x average depth, and from 1x to 2x observed still a big jump in imputation quality. That raises the question of whether the recommended 2x is where imputation performance "plateaus", and whether 3x or 4x could help even further (as the full data of Stuttgart of 11.1x suggest a higher plateau, Fig. S8).

This recommended coverage is an important question that directly impacts future study designs, as capture data for well-preserved samples can often cost-effectively be extended to reach such coverages by simply adding additional sequencing depth.

I would therefore suggest extending the imputation tests to 3x-5x for the Stuttgart capture data, as this seems to be directly possible with little effort within the current testing framework.

We thank the reviewer for the comment. We followed the reviewer's suggestion and additionally downsampled Stuttgart capture data to 3x, 4x and 5x, and subsequently imputed these data. Indeed, imputation accuracy for this genome seems to level off at around 4x. We updated Figure 5 to include these new results and edited the text accordingly.

A2)

Original Reviewer Comment:

C7) Giving switch error rates per cM would help (or the average cM of haplotype phased correctly), as they are more general and could even be compared across different SNP sets. It's also a property important for haplotype copying HMMs (see Biddanda et al. 2021, they looked into the expected switch rate in the Li and Stephens haplotype copying models in ancient samples).

Author's Response:

We disagree with the reviewer's assertion that the number of switches per cM is unaffected by the SNP set. This metric is influenced by the minor allele frequency, as a higher SNP density leads to an increase in rare variants, which are known to pose difficulties in phasing and are more prone to causing disruptions in runs of correctly phased heterozygous genotypes. In our study, we focussed on the switch error rate, as it is the most widely used metric in the literature to evaluate haplotype quality.

New Comment:

I did not want to assert that the number of switches per cM is unaffected by the SNP set - but merely that using switches per cM will allow comparing phasing quality in a metric that is key for downstream population genetic applications. E.g., the effects of different SNP sets mentioned in their rebuttal would become evident when comparing switch rates per cM (as this factors out the differences of SNP number).

Of course, the choice of metric is the authors and it is a valid argument to stay consistent with published literature.

However, I believe that it would be important to at least mention in a side sentence what the average genomic map distance is that can be phased correctly via imputation (on the 1000G SNP set) - as it has huge implications for downstream analysis such as LAI and ARG reconstruction whether average correctly phased segments are around 0.1, 1 or 10cM, say.

Computational imputation is the only viable way to obtain phased data for the vast majority of aDNA, and current downstream method developers often close their eyes and assume that computational phasing of aDNA is perfect and one obtains perfect long-range haplotypes (e.g. <https://doi.org/10.1101/2023.03.06.529121>).

This work could easily clarify the limits of the imputation-based phasing of ancient samples in terms of centimorgan so that for all population genetic method developers it becomes directly evident that assuming phasing on the order of several centimorgans is not viable for typical aDNA samples unless one uses trio-phasing. Talking about switch error rate per SNP number has the effect of hiding that from population genetic researchers.

We greatly appreciate the feedback provided by the reviewer and acknowledge the suggestion regarding the use of this phasing quality metrics. While we agree with this point for the most part, we regret to inform them that we are unable to incorporate this suggestion into the current version of our work. However, we would like to emphasize that we will take this suggestion into careful consideration in our future research on this topic.

A7)

L470-472: "Furthermore, using five genomes that were both obtained via in-solution capture and shotgun sequencing (>10x for the latter), we found that imputation performance of capture-sequenced data was highest at the capture sites for rare variants."

I think you mean "highest at the capture sites for common variants"? Or do you refer to the fact that rare variants "on" the capture panel were imputed better than rare variants outside of it?

We thank the reviewer for the remark, and we recognize that it is not clear. We refer to the fact that there is a larger difference between target and non-target sites at rare variants, compared to common variants. We edited the text to make it more clear as follows: "that imputation performance of capture-sequenced data was higher at the capture sites than outside of these and particularly for rare variants."

Reviewer #2 (Remarks to the Author):

I appreciate the authors largely addressing my previous comments. I am very happy with the current state of the manuscript.